# Plasma proteome profiling discovers novel proteins associated with non-alcoholic fatty liver disease

Lili Niu[1,2,†], Philipp E Geyer[1,2,†], Nicolai J Wewer Albrechtsen[1,2,3,4,5] (iD), Lise L Gluud[6,7], Alberto Santos[1], Sophia Doll[1,2], Peter V Treit[2], Jens J Holst[3,4] (iD), Filip K Knop[4,6,8], Tina Vilsbøll[6,8], Anders Junker[6,8], Stephan Sachs[9], Kerstin Stemmer[9], Timo D Müller[9], Matthias H Tschöp[9], Susanna M Hofmann[10,11,12] & Matthias Mann[1,2,*] (iD)

## Abstract

Non-alcoholic fatty liver disease (NAFLD) affects 25% of the population and can progress to cirrhosis with limited treatment options. As the liver secretes most of the blood plasma proteins, liver disease may affect the plasma proteome. Plasma proteome profiling of 48 patients with and without cirrhosis or NAFLD revealed six statistically significantly changing proteins (ALDOB, APOM, LGALS3BP, PIGR, VTN, and AFM), two of which are already linked to liver disease. Polymeric immunoglobulin receptor (PIGR) was significantly elevated in both cohorts by 170% in NAFLD and 298% in cirrhosis and was further validated in mouse models. Furthermore, a global correlation map of clinical and proteomic data strongly associated DPP4, ANPEP, TGFBI, PIGR, and APOE with NAFLD and cirrhosis. The prominent diabetic drug target DPP4 is an aminopeptidase like ANPEP, ENPEP, and LAP3, all of which are up-regulated in the human or mouse data. Furthermore, ANPEP and TGFBI have potential roles in extracellular matrix remodeling in fibrosis. Thus, plasma proteome profiling can identify potential biomarkers and drug targets in liver disease.

**Keywords** biomarker discovery; mass spectrometry; NAFLD; NASH; plasma proteome profiling
**Subject Categories** Genome-Scale & Integrative Biology; Molecular Biology of Disease; Post-translational Modifications, Proteolysis & Proteomics
**Mol Syst Biol. (2019) 15: e8793**

## Introduction

Non-alcoholic fatty liver disease (NAFLD) is the most common liver disease with an estimated prevalence of 25% in the general population of Western and Asian countries (Bellentani *et al*, 2010; Fan *et al*, 2017). NAFLD has become an enormous clinical and economic burden with annual medical costs of over $100 billion in the United States alone and is projected to keep growing in parallel with the increasing prevalence of obesity and type 2 diabetes (T2D) (Younossi *et al*, 2016). Unfortunately, progression is usually asymptomatic and only manifests when patients develop end-stage liver disease, which has limited treatment options. Although there is promising activity in the development of small-molecule drugs (Cassidy & Syed, 2016), there is currently no Food and Drug Administration (FDA)-approved drug with beneficial effects on clinical outcomes (Friedman *et al*, 2018).

NAFLD is defined as fat accumulation in the liver exceeding 5–10%, measured either by imaging methods or by liver histology after exclusion of other etiologies, for instance, heavy alcohol consumption and medication-induced steatosis (Kotronen & Yki-Jarvinen, 2008). NAFLD is further categorized histologically into simple steatosis and non-alcoholic steatohepatitis (NASH) with or without fibrosis. Up to 90% of patients with NAFLD have simple steatosis, with relatively benign prognosis and small risk of progression to advanced fibrosis and liver-related mortality, but an increased risk of cardiovascular events (Dyson *et al*, 2014; Singh *et al*, 2015). However, 10–30% of NAFLD patients have NASH, a more severe form of the disease with hepatocellular injury and hepatic

1 Novo Nordisk Foundation Center for Protein Research, Faculty of Health Sciences, University of Copenhagen, Copenhagen, Denmark
2 Department of Proteomics and Signal Transduction, Max Planck Institute of Biochemistry, Martinsried, Germany
3 Department of Biomedical Sciences, Faculty of Health and Medical Sciences, University of Copenhagen, Copenhagen, Denmark
4 Faculty of Health and Medical Sciences, Novo Nordisk Foundation Center for Basic Metabolic Research, University of Copenhagen, Copenhagen, Denmark
5 Department of Clinical Biochemistry, Rigshospitalet, University of Copenhagen, Copenhagen, Denmark
6 Department of Clinical Medicine, Faculty of Health and Medical Sciences, University of Copenhagen, Copenhagen, Denmark
7 Gastrounit, Medical Division, Hvidovre Hospital, University of Copenhagen, Hvidovre, Denmark
8 Clinical Metabolic Physiology, Steno Diabetes Center Copenhagen, Gentofte Hospital, Hellerup, Denmark
9 Helmholtz Diabetes Center at Helmholtz Centre Munich & Division of Metabolic Diseases, Institute for Diabetes and Obesity, Technische Universität München, Munich, Germany
10 Institute for Diabetes and Regeneration, Helmholtz Diabetes Center at Helmholtz Zentrum München, German Research Center for Environmental Health (GmbH), Neuherberg, Germany
11 German Center for Diabetes Research (DZD), Neuherberg, Germany
12 Medizinische Klinik und Poliklinik IV, Klinikum der LMU, München, Germany
*Corresponding author. Tel: +49 89 8578 2557; E-mail: mmann@biochem.mpg.de
†These authors contributed equally to this work

inflammation. NASH has a substantial risk of progression to advanced fibrosis and mortality (Singh et al, 2015). Early detection of NASH or fibrosis followed by lifestyle or pharmaceutical intervention would therefore be an ideal first step toward reducing future cirrhosis-related and cardiovascular deaths.

Liver biopsy remains the gold standard for diagnosing NASH and for staging the severity of fibrosis. However, there are inherent limitations of liver biopsy, as it is invasive, associated with complications such as bleeding, and suffers from sampling bias. Therefore, the field increasingly employs surrogate non-invasive techniques for the diagnosis of these conditions (Bril & Cusi, 2017). Radiographic techniques for the assessment of steatosis and liver stiffness include ultrasound, transient elastography (TE, FibroScan™; Echosens, Paris, France), multiparametric magnetic resonance imaging (MRI), magnetic resonance elastography (MRE), and magnetic resonance spectroscopy (MRS). TE with controlled attenuation parameter (CAP) accurately quantifies liver steatosis and stiffness in patients with NAFLD (Mikolasevic et al, 2016), but it does not have sufficient accuracy to discern between different stages of fibrosis (Chang et al, 2016). Furthermore, the use of TE in NAFLD has significant limitations in older patients and those with obesity (body mass index (BMI) > 35 kg/m$^2$) or type 2 diabetes as the rate of unreliable measurements is higher in these populations (Dyson et al, 2014). Even with a large-size (XL) probe for patients with obesity, there can still be a discordance in comparison with liver biopsy (Myers et al, 2012) and complementary methods would surely be beneficial.

Hepatic injury results in release of specific liver enzymes into the circulation, and these are routinely measured with blood tests. Classic markers for liver damage include alanine aminotransferase (ALT) and aspartate aminotransferase (AST). Elevated levels of these enzymes are often combined with patient data such as age and platelet count in NAFLD severity indices such as the fibrosis-4 (FIB-4) index (Sterling et al, 2006) and with biomarkers in the case of fatty liver index (Bedogni et al, 2006), NAFLD liver fat score (Kotronen et al, 2009), APRI, NAFLD fibrosis score (Angulo et al, 2007), and the Enhanced Liver Fibrosis panel (ELF) (Parkes et al, 2011). Among these tests, ELF is a commercial panel of three markers reflecting the process of extracellular matrix remodeling and fibrogenesis. This method has been found to be the most cost-effective non-invasive method in identifying patients with advanced liver fibrosis (stages 3 and 4) and has been recommended by NICE guideline as a blood test to screen for advanced liver fibrosis in adults (Glen et al, 2016). Most of these tests were developed more than a decade ago and comprise panels of simple clinical and laboratory variables. Furthermore, by their nature, these markers detect relatively late changes in liver pathology. Therefore, in the management of NAFLD, there is an unmet need to develop non-invasive or minimally invasive methods with better sensitivity and selectivity to detect NASH and fibrosis as well as to predict the progression of patients at risk.

After the fibrotic state, which is usually non-symptomatic, liver disease can further progress to cirrhosis and convert normal liver architecture into structurally abnormal liver nodules (Anthony et al, 1977). It is characterized by multiple severe physiological conditions, such as reduction in protein synthesis, abnormalities in the coagulation system, and portal hypertension. Different insults, for instance, viral infection, steatohepatitis, and autoimmune hepatitis, can all result in liver cirrhosis (Burt et al, 2015). Because of the serious prognostic implications of cirrhosis, developing risk markers or predictors of cirrhosis development is of utmost importance.

As a central secretory organ of the human body, the liver produces the majority of plasma proteins with a direct function in the circulation, which is why many of the classical biomarkers for liver dysfunction are in this category. By extension, liver disease is likely to affect the blood plasma proteome. Mass spectrometry (MS)-based proteomics is the technology of choice to analyze proteins in a systematic and systems-wide fashion. It has undergone persistent innovations in terms of sample preparation, instrumentation, acquisition methods, and computational analysis and has contributed to many breakthroughs in basic research over the last decades (Aebersold & Mann, 2003, 2016; Altelaar et al, 2013; Richards et al, 2015). Today, it clearly has the potential to facilitate disease-related biomarker discovery in an unbiased and non-hypothesis-driven manner (Geyer et al, 2017). Our group has recently developed an automated, rapid, and robust shotgun proteomics pipeline that allows the streamlined analysis of several hundred plasma proteins, a technology known as "plasma proteome profiling" (Geyer et al, 2016a). It requires only one microliter of plasma and features high reproducibility and low variability. So far we have applied this technology to study the effects of sustained weight loss on the human plasma proteome and to rigorously assess the quality of plasma samples (Geyer et al, 2016b; preprint: Geyer et al, 2018).

In this study, we employed and augmented plasma proteome profiling with a recently introduced data acquisition method termed "BoxCar" which covers the proteome with about tenfold higher dynamic range (Meier et al, 2018). We successfully applied this technology to identify new biomarker candidates for non-alcoholic fatty liver disease.

## Results

### Study design and assessment of the plasma proteome analysis

As the plasma proteomics pipeline has not yet been applied to liver disease, we first set out to study a phenotype for which the effects should be very drastic. Liver cirrhosis is a more severe condition than NAFLD and a common end-stage of most types of chronic liver diseases. As the liver has then undergone substantial changes in structure and function irrespective of disease etiologies, its analysis should form a basis for our general understanding of the effects of liver damage on the plasma proteome profile.

We first chose a cohort of ten non-diabetic patients with cirrhosis (cirrhosis-cohort) to compare them against ten age-, sex-, and BMI-matched healthy controls as well as against eight matched individuals with T2D and no liver disease (Junker et al, 2015; Fig 1A). However, our main goal was to investigate the changes in the plasma proteome in patients with NAFLD before progression to cirrhosis. As NAFLD is highly associated with obesity and T2D, we included both subtypes for separate comparisons (Junker et al, 2016; Fig 1A). The first cohort consisted of ten obese patients with NAFLD and normal glucose tolerance (NGT) (NAFLD-cohort 1), and the second included ten patients with both NAFLD and T2D (NAFLD-cohort 2). These cohorts were compared to the matched controls without NAFLD. The average duration of diabetes was 50 months. The cirrhosis and NAFLD studies had a total of 48

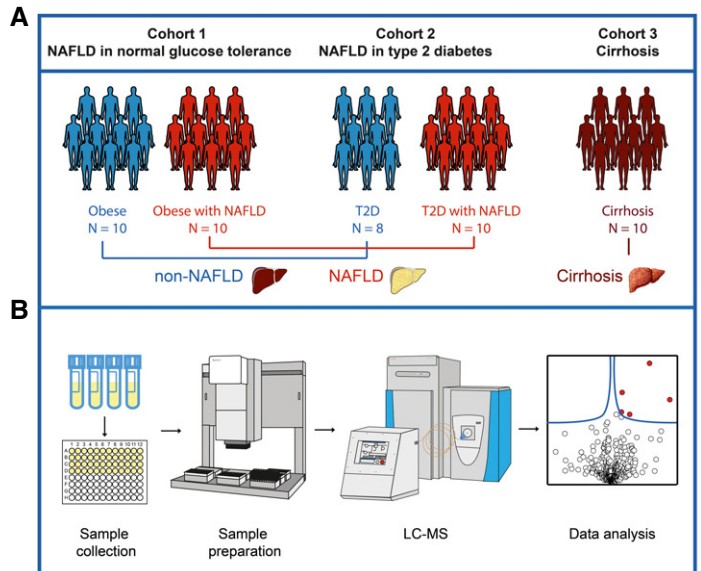

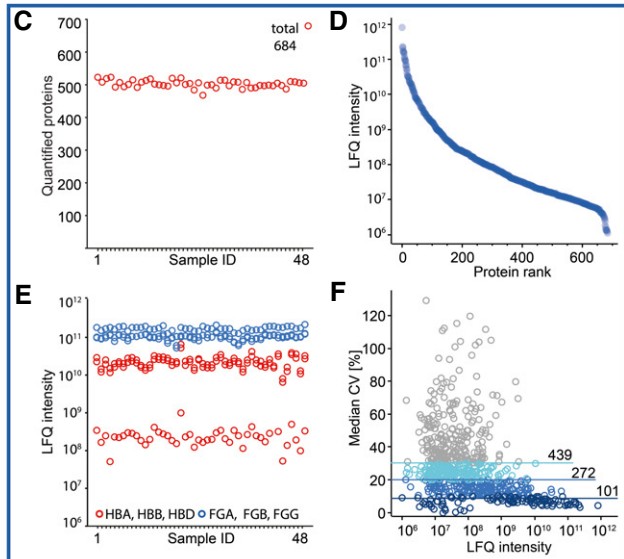

**Figure 1. Design and quality control of the human study.**

A   In total, 48 participants from three sub-studies of either NAFLD or cirrhosis with the indicated numbers of patients were included in this study.

B   Fasting plasma was collected and distributed into a 96-well plate for proteomic analysis. Proteins were denatured, reduced, alkylated, and digested using the automated plasma proteome profiling pipeline, and purified peptides were analyzed in triplicate measurements in a randomization manner by LC-MS/MS. The resulting 144 raw files were analyzed together with 168 library files by the MaxQuant and Perseus software programs.

C   Numbers of quantified proteins in the triplicate measurements.

D   Dynamic range of quantified proteins (LFQ, label-free quantitation values).

E   Assessment of study quality by analyzing erythrocyte-specific proteins (red circles) and coagulation markers (blue circles). HBA, HBB, HBD: hemoglobin subunits alpha, beta, delta; FGA, FGB, FGG: fibrinogen chains alpha, beta, gamma.

F   Assessment of quantitation accuracy of the LC-MS/MS instrumentation by the number of proteins with a coefficient of variation (CV) below 30, 20, or 10%, respectively, within three technical replicates.

participants (24 females) with a mean age of 57 years and a mean BMI of 28 kg/m$^2$ (Dataset EV1).

We improved our previously described plasma proteome profiling pipeline (Geyer *et al*, 2016a) by deep libraries in combination with a new acquisition method termed BoxCar (Meier *et al*, 2018). This method results in a tenfold increased dynamic range of peptide signals due to equalized filling pattern of the Orbitrap mass analyzer. The library consists of pooled and depleted plasma samples—including from patients with liver diseases (see Materials and Methods)—that were separated at the peptide level into 24 fractions. This deep plasma library consisted of in total 2,081 proteins. The study samples were then measured without depletion, but with BoxCar scans and in technical triplicates, resulting in about 150 LC-MS/MS datasets (Fig 1B). We quantified on average 503 proteins per individual (684 in total, of which 3 by single peptide) (Fig 1C), with MS signals that spanned an abundance range of six orders of magnitude (Fig 1C and D). To investigate the quality of the study samples in terms of consistency of collection and handling, we used our recently developed quality marker panels for coagulation and erythrocyte contamination (Geyer *et al*, 2016a; preprint: Geyer *et al*, 2018). No outliers of these marker indices were found based on global protein abundance profiles, indicating that the samples were of high quality and that changes in plasma proteins should be due to disease-related pathological disturbances (Fig 1E). We further analyzed the reproducibility of the measurements by calculating the coefficients of variation (CVs) within triplicate

measurements and found that 272 proteins had a median CV below 20% (Fig 1F).

## Plasma proteome profiling of patients with liver cirrhosis

We first compared the plasma proteome profiles of patients with liver cirrhosis to that of the two control groups (10 matched healthy controls and 8 individuals with only T2D but no liver disease). This revealed a dramatic shift in the quantitative proteome composition, reflected by 57 significantly differentially abundant proteins (Student's *t*-test, *P* < 0.01, permutation-based FDR < 0.05), of which 31 proteins were up-regulated and 26 down-regulated (Fig 2A).

The down-regulated proteins include several classes with a direct function in the bloodstream, and almost all of them are produced in the liver. One of the classes contained proteins regulating blood coagulation and fibrinolysis, for instance, prothrombin (F2), vitamin K-dependent protein C and S (PROC, PROS1), alpha-2-antiplasmin (SERPINF2), antithrombin-III (SERPINC1), kallikrein (KLKB1), heparin cofactor II (SERPIND1), and carboxypeptidase (CPB2) (Fig 2B). This reflects the central role of the liver in hemostasis, in which most clotting and anticoagulation factors are synthesized by parenchymal cells that are also involved in the clearance of activated products (Palta *et al*, 2014). The prominent 12–30% abundance changes of these proteins in the plasma proteome (Fig 2C) are likely to underlie the well-known disturbances of the coagulation system associated with cirrhosis and affecting the prolonged

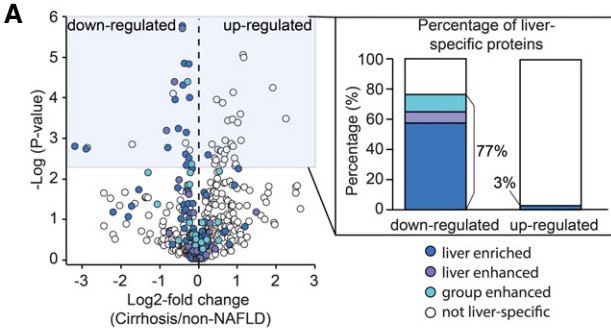

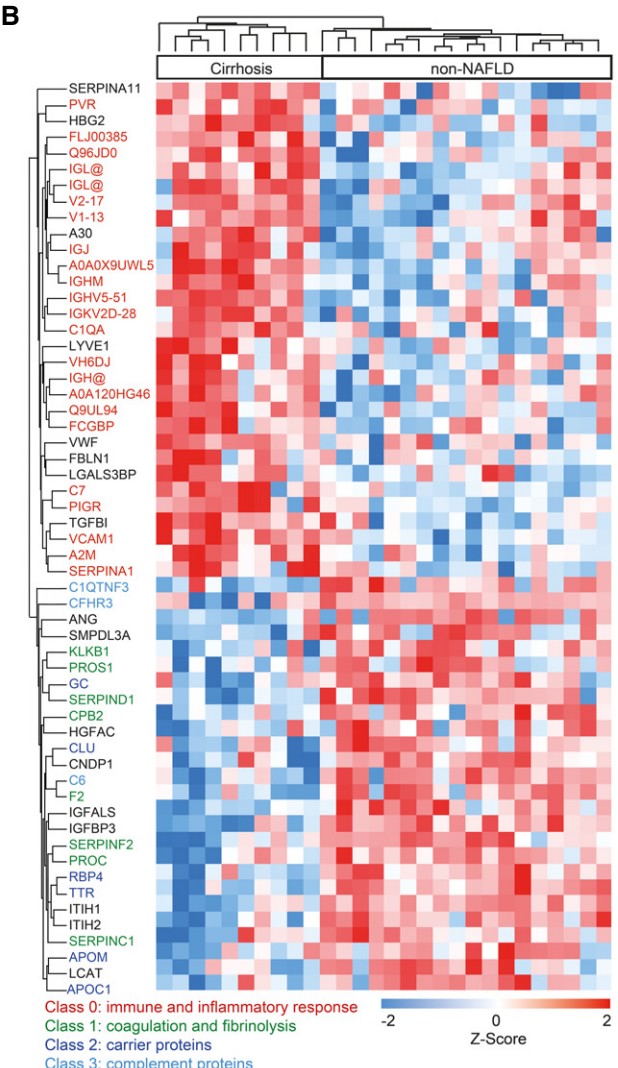

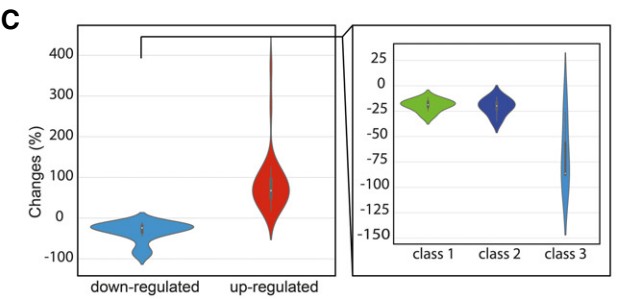

Figure 2. Reduced protein production and increased immunological response in cirrhotic liver.

A  Volcano plot of statistical significance against log2-fold change between the cirrhosis group (N = 10) and non-NAFLD group (N = 18), showing significantly differentially expressed proteins shaded in blue and down-regulated liver-specific proteins color-coded according to the classification of Human Protein Atlas (HPA). Significance was defined by independent two-sample t-test (two-sided) corrected by permutation-based FDR of 0.05. The percentage of down- and up-regulated "liver-specific" proteins is indicated.

B  Hierarchical clustering of significantly expressed proteins between the cirrhosis group and non-NAFLD group. Intensities of proteins were log2-transformed and Z-scored to normalize across individuals. Proteins involved in different biological processes or belonging to different classes are indicated by color.

C  Violin plot of mean fold changes for down- and up-regulated proteins. The fold changes of down-regulated proteins were further calculated separately for the three protein classes indicated in panel B.

bleeding time and increased thrombosis risk in these patients. A second class consisted of carrier proteins such as apolipoprotein M (APOM), apolipoprotein C1 (APOC1), and clusterin (CLU), which play important roles in cholesterol metabolism and are likely to be associated with cirrhosis-related dyslipidemia due to reduced liver biosynthesis capacity (Chrostek et al, 2014). In addition, we found important changes in proteins associated with hormone and vitamin transportation such as retinol-binding protein (RBP4), transthyretin (TTR), and vitamin D-binding protein (GC). A final class of proteins are components of the complement system such as complement component 6 (C6), complement factor H-related protein 3 (CFHR3), and complement C1q tumor necrosis factor-related protein 3 (C1QTNF3) (Fig 2B). Members of the latter two classes were down-regulated substantially, reflecting severe damage to liver function in cirrhosis (Dataset EV1).

The majority of the up-regulated proteins are involved in immune system regulation and inflammation such as complement component C7 (C7), immunoglobulin chains (immunoglobulin heavy variable 5–51, immunoglobulin J chain, and immunoglobulin heavy constant mu), polymeric immunoglobulin receptor (PIGR), vascular cell adhesion protein 1 (VCAM1), alpha-2 macroglobulin (A2M), and alpha-1-antitrypsin (SERPINA1). Among these proteins, A2M is an established marker of liver fibrosis (Naveau et al, 1994) and already incorporated into biomarker blood tests, for example, the SteatoTest (Poynard et al, 2005) and FibroTest (Imbert-Bismut et al, 2001). In our study, A2M showed a 54% up-regulation in patients with cirrhosis.

To investigate the tissue origin of the up-regulated and down-regulated proteins, we used the data and annotation of the Human Protein Atlas (HPA) (Kampf et al, 2014). Of note, 77% of the down-regulated proteins are classified as "liver-specific" with different degrees of enrichment in the liver relative to the rest of the body (see Materials and Methods). In stark contrast, only one of the 31 up-regulated proteins was annotated as liver-specific. This indicates that down-regulation of liver proteins is a reflection of impaired hepatic protein synthesis and secretion in patients with cirrhosis, whereas up-regulation represents a systemic response of the body to cirrhosis (Fig 2A and B). We conclude that liver cirrhosis is clearly reflected in the plasma proteome profiles of patients through changes in abundance of proteins originated from the diseased

organ as well as specific biological processes that are directly related to the disease phenotype.

## Plasma proteins highly associated with NAFLD

Having revealed proteome changes in plasma of patients with cirrhosis, we next investigated if the same proteins were affected in milder forms of liver disease. Our long-term motivation is to help improve detection of NAFLD, which has proved to be challenging, especially in relation to the identification of the subsets of patients who will progress. NAFLD is very heterogeneous and is highly associated with obesity and T2D. The prevalence of ultrasonographic NAFLD in patients with T2D reaches 70% (Leite *et al*, 2009), much higher than the 25% of the general population with this diagnosis (Bellentani *et al*, 2010). The presence of T2D is an independent predictor of NASH and advanced fibrosis in NAFLD and vice versa (Williams *et al*, 2013; Bazick *et al*, 2015). Based on these findings, we decided to analyze our two NAFLD cohorts for common and potentially separate protein markers.

Our analyses showed a 186% increase of the polymeric immunoglobulin receptor (PIGR), a 341% increase of fructose-bisphosphate aldolase B (ALDOB), and a 22% increase of vitronectin (VTN) in NAFLD patients with NGT (NAFLD-cohort 1) compared with healthy controls ($P < 0.001$, Fig 3A and B). Immunoglobulin heavy constant delta (IGHD) was down-regulated in NAFLD patients by 67%; however, it was not significant in an up-front one-way ANOVA across all five experimental groups and was therefore excluded from this panel (Table EV1).

Previous studies have found up-regulation of immunoglobulin A in patients with NAFLD (Inamine & Schnabl, 2018), and this has been proposed as a biomarker of changes in the gut microbiota. No previous evidence has associated NAFLD with immunoglobulin G (IgG) or M (IgM), and none of the included patients had serum immunoglobulins outside the normal range. In accordance with the increased levels of PIGR in patients with NAFLD, we also observed up-regulated immunoglobulin chains in cirrhosis, among which most are derived from IgA, IgM, and IgG. These indicate an elevated immunological response in patients with NAFLD and cirrhosis.

In the NAFLD-cohort 2, we found statistically significant increases in the levels of PIGR by 157%, galectin-3 binding protein (LGALS3BP) by 102%, and AFM by 58% (Fig 3C and D). APOM was significantly decreased by 25%. Like IGHD, IgGFc binding protein (FCGBP) was also statistically different between disease and control groups but was excluded from the panel due to insignificance in the one-way ANOVA (Table EV1). This associates a panel of six proteins, ALDOB, PIGR, VTN, LGALS3BP, AFM, and APOM, with NAFLD. At least one of them, PIGR, associated with NAFLD independently of the glucose tolerance.

AFM and LGALS3BP have already been suggested as potential markers for NAFLD (Bell *et al*, 2010; Wood *et al*, 2017), providing a positive control. AFM is a human vitamin E-binding protein in plasma that is primarily expressed in the liver and secreted into the circulation. It is strongly associated with components of the metabolic syndrome (Dieplinger & Dieplinger, 2015), NAFLD, and alcoholic liver disease (ALD) (Bell *et al*, 2010; Liu *et al*, 2011; Neuman *et al*, 2014). LGALS3BP, which was up-regulated in NAFLD and T2D, is a candidate biomarker for hepatitis C-related fibrosis and cirrhosis (Cheung *et al*, 2010). Interestingly, galectin-3, the ligand of

LGALS3BP, is likewise associated with fibrosis in diverse tissues, for instance, kidney, lung, liver, and heart (Li *et al*, 2014). Its critical role in fibrosis has led to ongoing studies to develop galectin-3 targeted anti-fibrotic drugs. An inhibitor targeting galectin-3, GR-MD-02, is currently entering phase III trials for NASH and thus may lead to a first therapy for the treatment of fatty liver disease with cirrhosis (Henderson *et al*, 2006; Harrison *et al*, 2016). The relation between galectin-3 and LGALS3BP in the context of fibrogenesis remains to be investigated.

We next asked if any of the NAFLD-associated plasma proteins were also altered in the plasma proteome of cirrhosis patients. This was the case for three of the six proteins, which changed also in NAFLD: PIGR (298%), LGALS3BP (170%), and APOM (21%) (Fig EV1A and B). Interestingly, the level of PIGR was up-regulated in all three cohorts and 128% higher in the cirrhosis group compared to NAFLD, consistent with the notion that increased PIGR levels may be a novel marker of disease progression (Fig EV1C). LGALS3BP is also more strongly up-regulated in cirrhosis. The fact that ALDOB, the protein with the largest fold change in NAFLD, showed no difference in cirrhosis may be due to the impaired protein synthesis and secretion that was apparent in the cirrhotic liver (Fig EV1D).

## Global correlation map reveals a panel of five plasma proteins correlated with liver enzymes

Routine blood tests provide evidence for abnormalities in the liver, for example, inflammation and hepatocyte cell death. The most commonly tested liver enzymes are ALT, AST, alkaline phosphatase (ALP), and gamma-glutamyl transferase (GGT). Note that levels of these enzymes do not necessarily reflect liver function but rather liver damage. To evaluate if the plasma concentration of liver enzymes was associated with other proteins, we employed global correlation analysis of the plasma proteome (Wewer Albrechtsen *et al*, 2018). Filtering for a quantitative data completeness for at least 70%, pairwise correlation of the liver enzymes and the quantified proteins resulted in a data matrix with 431 protein levels and 21 clinical parameters. Correlating all variables with each other followed by hierarchical clustering led to a global correlation map containing about 100,000 Pearson correlation coefficients, with clearly apparent clusters of co-varying proteins and variables (Fig 4A). Strikingly, one of these groups contained all the above-mentioned liver enzymes, as well as five further proteins quantified by plasma proteome profiling (Fig 4B). One of these is PIGR, which we had identified as a protein with the potential to discriminate the non-NAFLD and the NAFLD groups. The other four are also highly relevant for liver disease: APOE (apolipoprotein E), dipeptidyl peptidase-4 (DPP4), TGFBI (transforming growth factor-beta-induced protein ig-h3), and aminopeptidase N (ANPEP). Individually correlating each of the four liver enzymes with the plasma proteome confirmed very high statistical significance, such as that of ALT to ANPEP (Pearson correlation 0.69; $P < 10^{-7}$; Fig EV2A).

After identifying proteins correlating with the established liver damage marker ALT by the global correlation map, we asked if we could reproduce these correlations in our very recently published study on the effect of bariatric surgery in morbidly obese individuals on the plasma proteome (Wewer Albrechtsen *et al*, 2018).

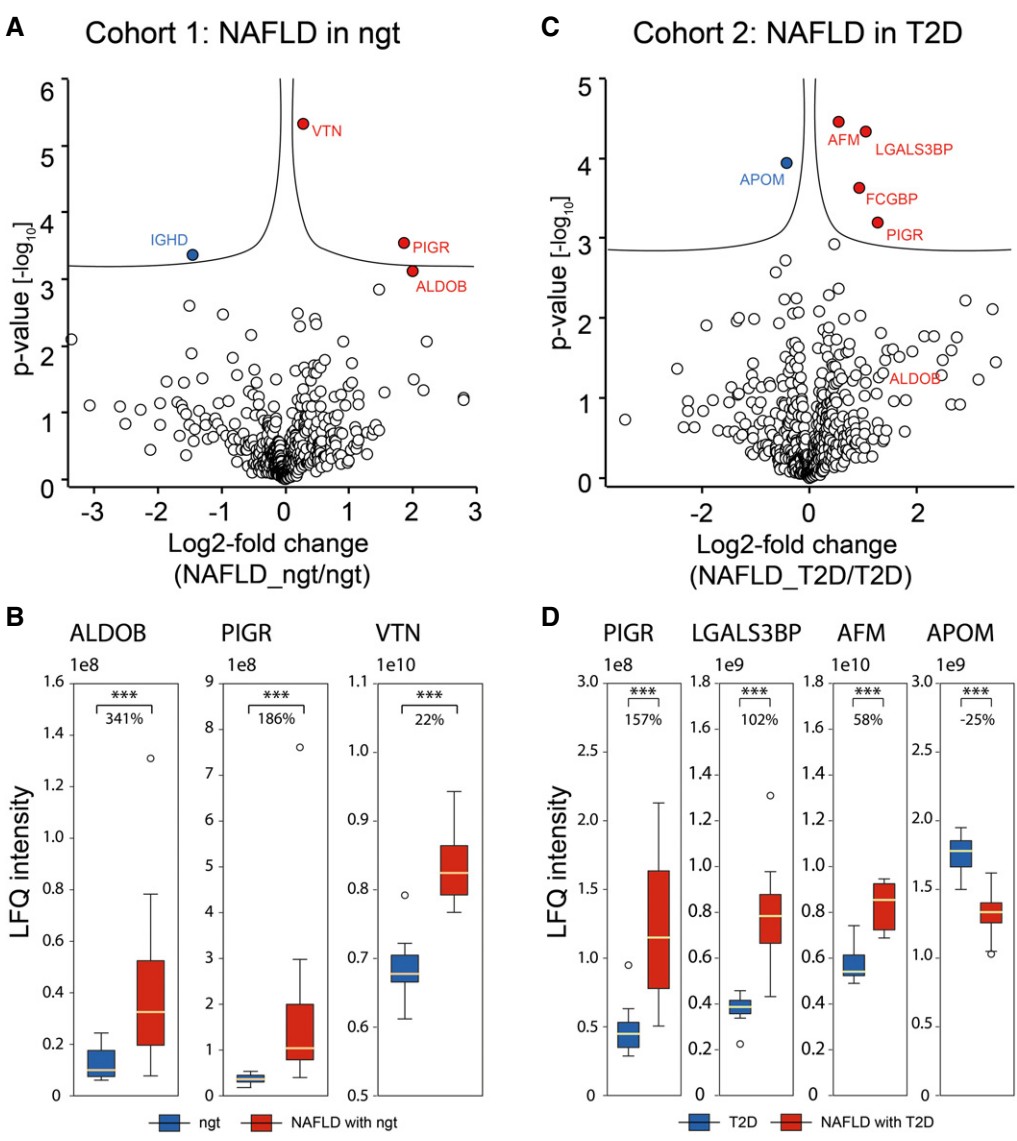

**Figure 3. A panel of proteins strongly associated with NAFLD in human cohorts.**

A   Volcano plot of statistical significance against log2-fold change between NAFLD (N = 10) and controls (N = 10) in NAFLD subtype 1: NAFLD in normal glucose tolerance. Significance is controlled by P-value (independent two-sample t-test, two-sided) and minimum fold change (s0 parameter in Perseus) indicated by the cutoff curve, demonstrating significantly up-regulation of PIGR, ALDOB, and VTN.

B   Box-and-whisker plot showing the distribution of mass spectrometric intensity values of three proteins in the first NAFLD cohort with median fold changes. The yellow line is the median, the top and the bottom of the box represent the upper and lower quartile values of the data and the whiskers represent the upper and lower limits for considering outliers (Q3+1.5*IQR, Q1-1.5*IQR) where IQR is the interquartile range (Q3–Q1). ***, P < 0.001 (independent two-sample t-test, two-sided). Number of replicates is defined in panel (A).

C   Volcano plot of statistical significance against log2-fold change between NAFLD (N = 8) and controls in NAFLD (N = 10) subtype 2: NAFLD in T2D, showing that AFM, LGALS3BP, and PIGR are significantly up-regulated and APOM significantly down-regulated.

D   Box-and-whisker plot showing the distribution of mass spectrometric intensity values of four proteins in the second NAFLD cohort with median fold changes. Representation of boxes and whiskers is defined as in panel (B). Number of replicates is defined in panel (C).

Correlating protein levels with clinically determined ALT levels across 175 plasma samples confirmed that ALDOB, PIGR, and ANPEP are statistically significantly correlated, whereas TGFBI is very close to the threshold (Fig EV3; Dataset EV2).

The global correlation map also showed a significant association between DPP4 and liver enzymes. Although DPP4 was elevated by 29% in NAFLD patients compared to non-NAFLD controls, this increase was not statistically significant after correction for multiple hypothesis testing (P = 0.02). Studies have shown that increased hepatic expression of DPP4 is associated with NAFLD (Balaban et al, 2007; Miyazaki et al, 2012; Itou et al, 2013). Hepatocyte-specific DPP4 overexpression in mice increases body fat and promotes hepatic steatosis, suggesting that this association is causative (Baumeier et al, 2017). Taken together with our proteome

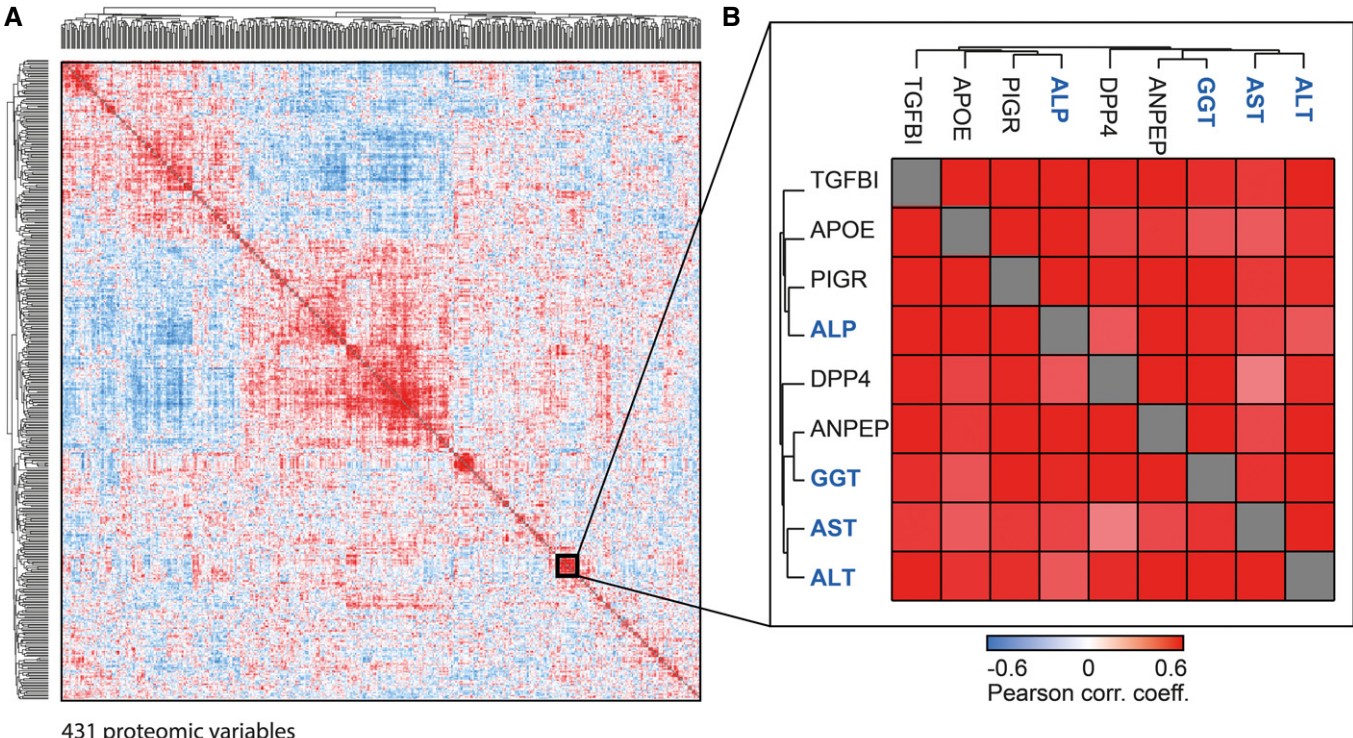

431 proteomic variables
21  clinical parameters

**Figure 4.  Global correlation map of the plasma proteome and clinical variables in human cohorts.**

A  Pairwise correlation of proteins and clinical variables over the 48 study participants, resulting in a matrix of correlation coefficients where each variable is compared to all others. Variables with a high positive correlation to each other will cluster together in groups of red rectangles (high correlations). Negative correlation is indicated in blue patches.

B  The magnified area highlights a cluster of variables that contains the four main clinical measurements for liver diseases (blue names) as well as five proteins, which were quantified by plasma proteome profiling (black gene names).

profiling results, levels of soluble DPP4 in plasma may represent an even stronger candidate as an indicator of liver damage. Biologically, it is a serine protease targeting incretins such as glucagon-like peptide-1 (GLP-1) and glucose-dependent insulinotropic peptide (GIP). DPP4 inhibitors are widely used as blood glucose-lowering agents in T2D (Deacon, 2018). DPP4 also plays a role in the degradation of extracellular matrix, the imbalance of which is a hallmark of liver fibrosis (Itou *et al*, 2013).

Interestingly, the correlation analysis showed a connection between liver enzymes, TGFBI and ANPEP, which are also involved in the remodeling of extracellular matrix (Bieche *et al*, 2005). However, nothing is known about a possible role of TFGBI and ANPEP in liver disease. The up-regulated levels of the extracellular matrix protein TGFBI that we observed in these patients could reflect early scar tissue formation in the liver, which in turn may induce increased levels of DPP4 and ANPEP.

**Investigation of the plasma proteome in a mouse model of high-fat diet-induced NAFLD**

As a MS-based method, plasma proteome profiling does not rely on specific protein epitopes (unlike most antibodies) and can

generically be applied across species to determine to which degree the biological responses are similar between them. There are well-established NAFLD mouse models that are generated by high-fat diet (HFD), and these are commonly used to study the effect of incretin agonist treatment for diet-induced obesity. We took advantage of a NAFLD mouse cohort designed to determine the effects of incretin agonists—alone and in combination—on improving the metabolic phenotype. It consisted of mice with mild and severe NAFLD induced by high-fat diet for either up to two months or more than six months (Fig 5A). The mice with mild NAFLD were treated with vehicle for 21 days, while the mice with severe NAFLD were randomly divided into four groups, treated with either vehicle, GIP receptor agonist, GLP-1 receptor agonist, or GIP/GLP-1 co-receptor agonist for 15 days while maintaining a high-fat diet. These mono and dual receptor agonists have previously been reported to reduce body weight and hepatic steatosis. The unimolecular dual GIP/GLP-1 receptor agonist has shown superior efficacy relative to GIP or GLP-1 receptor agonists in the past (Frias *et al*, 2017; Jall *et al*, 2017). Therefore, in our cohorts it is expected to have a larger effect than the two other treatments, providing four groups of disease trajectories of mice with different severity of NAFLD. This study design allows us to investigate the potential of our newly identified

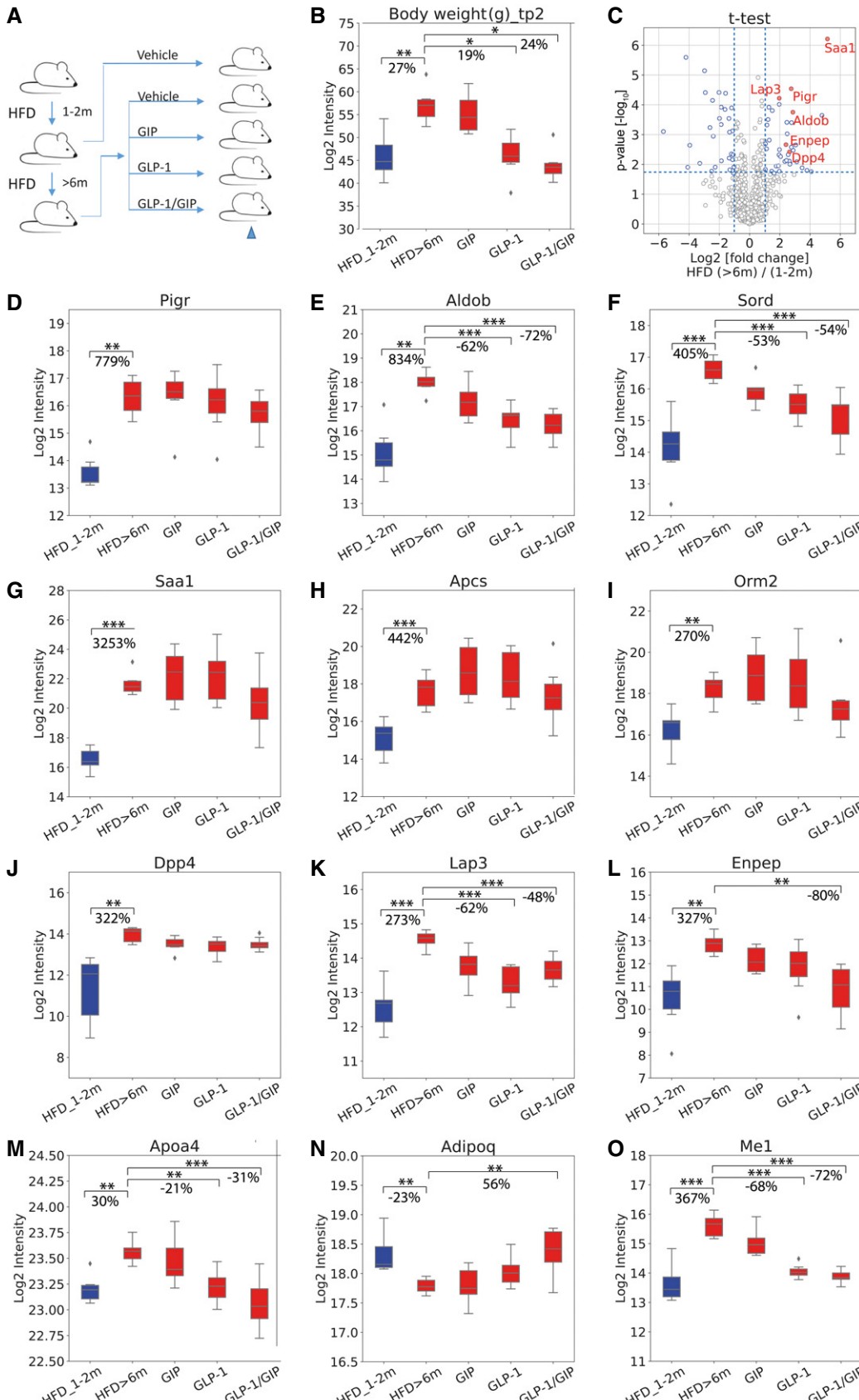

**Figure 5.**

**Figure 5. Plasma proteome changes in a HFD-induced NAFLD mouse model.**

A    Mouse cohort design.

B    Box-and-whisker plot showing the distribution of log2-intensity values of body weight across five groups: HFD_1-2 m (N = 6), HFD > 6 m (N = 6), GIP (N = 7), GLP-1 (N = 8), and GLP-1/GIP (N = 6).

C    Volcano plot of statistical significance against log2-fold change between mice on > 6 months of HFD and mice on 1–2 months of HFD. Significance is controlled by FDR-corrected *P*-value and minimum log2-fold change of 1 indicated by the blue-dotted line, demonstrating that Saa1, Pigr, Aldob, Lap3, Enpep, and Dpp4 are significantly up-regulated.

D–O    Box-and-whisker plot showing the distribution of log2-intensity values of statistical significantly regulated proteins across five groups. Number of replicates is defined in panel (B). The yellow line is the median, the top and the bottom of the box represent the upper and lower quartile values of the data and the whiskers represent the upper and lower limits for considering outliers (Q3+1.5*IQR, Q1-1.5*IQR) where IQR is the interquartile range (Q3–Q1).

Data information: Significance was defined by independent *t*-test (two-sided) followed by Benjamini–Hochberg correction for multiple hypothesis testing with a significance level of **P* < 0.05, ***P* < 0.01, and ****P* < 0.001.

marker candidates to monitor treatment effects of this co-agonist in NAFLD.

Mice with severe NAFLD exhibited a 27% increase in body weight compared to mice with mild NAFLD. Upon treatment with GLP-1 mono-agonist or GIP/GLP-1 co-agonist, they lost 19% or 24% of body weight, respectively (Fig 5B). There was no significant weight loss upon GIP agonist treatment alone. *T*-test analysis of the quantitative changes in the plasma proteome between severe and mild NAFLD mice resulted in 71 significant hits—40 up-regulated and 31 down-regulated (Fig 5C–O). Pigr and Aldob, which we had found to be associated with NAFLD in the human cohorts, were also highly significantly up-regulated upon progression in mice (Fig 5C). Median plasma levels of Pigr in severe NAFLD mice were 780% higher than those in mild NAFLD (Fig 5D), an even greater change than in the human cohorts. While Pigr levels decreased upon treatment, especially in the GLP-1 and combination treatment, these changes were not significant. Like Pigr, plasma levels of Aldob are drastically increased in NAFLD compared to mild NAFLD (830%, Fig 5E). These increases were mostly reversed upon treatment by GLP-1 and GIP/GLP-1 receptor agonist (down-regulation by 62 and 72%, respectively). Sord (sorbitol dehydrogenase), another enzyme involved in fructose metabolism, followed a very similar pattern trajectory (Fig 5F).

The plasma levels of three proteins that are involved in the acute phase reaction increased strongly upon progression from mild to severe NAFLD: Saa1 (serum amyloid A-1 protein) by 3,250%, Apcs (serum amyloid P-component) by 440%, and Orm2 (alpha-1-acid glycoprotein 2) by 270% (Fig 5G–I). In a weight loss study in humans, we had previously classified these proteins as members of a plasma protein panel indicating increased systemic inflammation (Geyer *et al*, 2016b). Apcs and Orm2 are primarily expressed in the liver, and Saa1, which is highly up-regulated in response to inflammation and tissue injury, is expressed in both liver and adipose tissue according to the Human Protein Atlas. Given that inflammation of visceral adipose tissue contributes to the development of insulin resistance and steatohepatitis, increased inflammatory factors in the blood in our experimental setup could originate from both liver and adipose tissue, specifically hepatocytes, adipocytes, and resident or recruited macrophages.

Plasma levels of Dpp4 increased by 322% in abundance in severe NAFLD (Fig 5J) compared to mild NAFLD. This could be due to tissue leakage or proactive secretion mediating tissue crosstalk via circulating factors. A recent study showed that obesity in mice stimulates hepatocytes to synthesize and secrete Dpp4, which promotes visceral adipose tissue inflammation and insulin resistance (Ghorpade *et al*, 2018). Interestingly, along with Dpp4, two other aminopeptidases—Lap3 (leucine aminopeptidase 3) and Enpep (glutamyl aminopeptidase)—also increased in severe NAFLD compared to mild NAFLD, by 273 and 327%, respectively, followed by a decrease upon treatment (Fig 5K and L). Protein–protein correlation analysis of the entire dataset of 82 mice in our experimental setup (Dataset EV3) reveals that Lap3, Enpep, and Me1 (NADP-dependent malic enzyme) correlate with Pigr, Aldob, and Dpp4 (Fig EV4). This is consistent with our human study, where PIGR, DPP4, and ANPEP co-vary with the four classic liver enzymes.

We also found that Apoa4, a major component of high-density lipoprotein and chylomicrons, increased by 30% upon progressing from mild to severe NAFLD. This was partially reversed upon GLP-1 receptor agonist and GIP/GLP co-agonist treatment (21 and 31% decrease, respectively; Fig 5M). Adiponectin negatively correlates with body weight and its plasma concentration is reduced in patients with NASH (Balmer *et al*, 2010). We found adiponectin levels decreased by 23% in severe NAFLD compared to mild NAFLD, followed by increased levels in the three treatment groups compared to the non-treatment group (Fig 5N).

## Discussion

Currently established protocols in clinical practice for the diagnosis and follow-up of NAFLD have certain limitations; for instance, they may not be sufficiently sensitive at early disease stages. MS-based proteomics technology holds great potential in generating novel insights into disease mechanism and discovering new biomarkers. To identify novel proteins associated with NAFLD and to understand the effect of liver cirrhosis on the plasma proteome, we here analyzed plasma samples of 48 participants with our streamlined plasma proteomics workflow, enhanced with a novel MS-acquisition method featuring high sensitivity.

Our results revealed clinically interesting proteome changes in NAFLD and in cirrhosis, where we found dysregulation of proteins associated with thrombosis and homeostasis. These findings are the proteomic reflection of the hemostatic complications of liver disease, in particular increased risk of bleeding and venous thromboembolism in patients with cirrhosis (Yang *et al*, 2014). Vitamin A and D deficiency is a common feature in chronic liver disease. We found proteins involved in hormone and vitamin transportation to

be altered in patients with cirrhosis, for example, retinol-binding protein (RBP4) and transthyretin (TTR). Reduced RBP4 levels are in concordance with a previous study (Bahr *et al*, 2009). Both RBP4 and TTR are primarily produced in hepatocytes and transport retinol to peripheral tissues. In the liver, hepatocytes secret RBP4, to enable retinoid storage in the hepatic stellate cells (HSC), accounting for as much as 50–60% of the total retinoid present in the body. The insufficient levels of RBP4 observed by proteomics might therefore contribute to vitamin A deficiency.

Among the interesting similarities between patients with cirrhosis and patients with NAFLD was PIGR, a little studied protein produced in the GI tract and endothelial cells but also in the liver. Strikingly, PIGR increased in abundance in all our three cohorts in line with the severity of liver damage and the up-regulation is most dramatic in cirrhosis. We further validated this novel finding in a mouse model of NAFLD induced by high-fat diet. PIGR is a receptor, which mediates transcytosis of immunoglobulins from the basolateral to the apical surface of the epithelia, facilitating the secretion of IgA and IgM. PIGR levels also co-varied with AST, ALT, ALP, and GGT, four clinical liver markers. The inter-individual variation in plasma PIGR levels in controls was relatively small, much lower than that in the NAFLD and cirrhosis cohorts. This finding was also observed in patients with T2D with no liver damage, thus making PIGR an interesting biomarker candidate for the inclusion in liver damage tests.

The plasma proteome changed much less in NAFLD than in cirrhosis and globally the plasma proteome profiles had few significant outliers, both in the cohort with normal glucose tolerance and in the T2D cohort. This presumably reflects the resilience and regenerative capacity of the liver and is also in line with the fact that NAFLD or early cirrhosis is often asymptomatic and clinically difficult to detect. Circulating RBP4 levels are among the changes that have been controversially discussed in the literature, with some studies finding higher levels in NAFLD (Terra *et al*, 2013) and some reporting unchanged levels (Zhou *et al*, 2017). This parallels our plasma proteome data, in which RBP4 was clearly significant in cirrhosis but not in the NAFLD sub-groups.

There is increasing awareness of the importance of screening for the presence of NAFLD, especially in patients with T2D. However, this requires markers or marker panels that are specific to NAFLD and ideally also associated with its progression to fibrosis and cirrhosis. Our plasma proteome profiling experiments of three cohorts provide a step in this direction. We identified a panel of six proteins in the two NAFLD subtypes: three in NAFLD without T2D (PIGR, ALDOB, and VTN) and four in NAFLD with T2D (PIGR, LGALS3BP, AFM, and APOM). Of these, AFM and LGALS3BP have been reported as potential markers for NAFLD (Bell *et al*, 2010; Wood *et al*, 2017). AFM has been closely linked to metabolic syndrome, insulin resistance, NAFLD, and alcoholic liver disease (Bell *et al*, 2010; Liu *et al*, 2011; Neuman *et al*, 2014; Kollerits *et al*, 2017). LGALS3BP has already been used to build multi-component classifiers for the prediction of NAFLD (Wood *et al*, 2017) and fibrosis in patients with hepatitis C infection (Cheung *et al*, 2010). The elevated levels of LGALS3BP and vitronectin (VTN), another extracellular matrix (ECM) protein, are likely a reflection of remodeling of the ECM in liver disease.

ALDOB is a glycolytic enzyme in the fructose catabolism pathway that also plays a role in gluconeogenesis and lipogenesis. Evidence from both animal studies and human studies suggests that dietary added fructose intake when consumed in excess is a principal driver of NAFLD and its deleterious consequences (DiNicolantonio *et al*, 2017). We also observed increased plasma ALDOB levels in the mouse NAFLD model. This increase may be a consequence of cell damage, as hepatocytes inflamed due to excessive fat accumulation may leak out ALDOB. Since that is visible already in NAFLD, it may be more sensitive than the usually measured liver enzymes AST and ALT.

In our analysis, we also correlated all proteomics and clinical variables with each other to identify proteins correlating with known markers for liver diseases under pathophysiological conditions. In this global correlation analysis, involving close to 100,000 individual correlation coefficients, proteins that are tightly co-regulated likely share a common origin or (patho)physiological pathway. Remarkably, in this unbiased analysis, four clinical markers used in liver disease—ALT, AST, ALP, and GGT—clustered into a single group together with a panel of five proteins of special interest—PIGR, DPP4, ANPEP, TGFBI, and APOE. Among these proteins are two enzymes involved in degradation of the ECM (ANPEP, DPP4) and an ECM protein (TGFBI). This may again reflect the remodeling of the ECM in the liver, which is part of the progression to liver fibrosis. Even more interesting, DPP4, which proteolytically cleaves GLP-1 and GIP, has been reported to increase in both plasma and liver tissue of NAFLD patients (Miyazaki *et al*, 2012; Anoop *et al*, 2017). DPP4 inhibitors are also widely used diabetes drugs worldwide. Interestingly, there are currently clinical trials evaluating the efficacy and safety of DPP4 inhibitor sitagliptin for the treatment of NAFLD (Fukuhara *et al*, 2014) and in steatosis irrespective of diabetes in NASH (Alam *et al*, 2018). Taken together with our proteomic data, it would be interesting to further investigate the potential of circulating DPP4 levels to serve as prognostic marker for both T2D and NAFLD. The same may apply for the other four proteins in this five-protein panel as they also correlate very well with liver enzymes of hepatic damage. Remarkably, we also observed this close correlation between DPP4 and PIGR in our mouse dataset, where we found that two additional aminopeptidases, Lap3 and Enpep, similarly were increased under NAFLD. Together, our analysis points to an important relationship between aminopeptidases and NAFLD.

In conclusion, in this study we found changes in the plasma proteome of patients with cirrhosis and patients with NAFLD that are clearly linked to the underlying disease manifestations and clinical observations. PIGR and ALDOB are in our panel of six proteins significantly associated with NAFLD and were also validated in a mouse NAFLD cohort making them interesting candidates for follow-up studies. PIGR is a particularly promising protein as its association with NAFLD and liver cirrhosis is novel, and its levels in plasma are highly correlated with DPP4, a widely used drug target in the treatment of T2D. This correlation holds true in both human and mouse cohorts. Blocking the enzymatic function of the other aminopeptidases may have effects on the fibrotic process in NAFLD progression. To evaluate the potential and specificity of the above proteins to be developed or incorporated into liver disease panels, we plan to perform plasma proteome profiling on larger and more fine-grained cohorts.

   

# Materials and Methods

**Reagents and Tools table**

| Reagent/Resource | Reference or Source | Identifier or Catalog Number |
|---|---|---|
| **Experimental Models** | | |
| Blood plasma samples (H. sapiens) | (Junker *et al*, 2015, 2016) | N/A |
| Blood plasma samples (M. musculus) | This study | N/A |
| **Chemicals, enzymes and other reagents** | | |
| Isopropanol | Sigma Aldrich/Merck | Cat # 67-63-0 |
| Formic acid | Sigma Aldrich/Merck | Cat # 64-18-6 |
| Acetonitrile | Sigma Aldrich/Merck | Cat # 75-05-8 |
| Pierce Dimethylsulfoxide (DMSO), LC-MS Grade | Thermos Fisher Scientific | 85190 |
| Water, Optima™ LC/MS Grade | Fisher Chemical | Cat # W64 |
| 25% LC-MS grade ammonia | Merck Millipore | Cat # 533003 |
| Trifluoroacetic acid | Sigma Aldrich/Merck | Cat # 76-05-1 |
| Multiple Affinity Removal Column Human 6 | Agilent | 5188-5341 |
| Seppro Protein Depletion | Sigma Aldrich | SEP010 |
| **Software** | | |
| MaxQuant (1.5.8.6) | https://maxquant.org/ | N/A |
| Perseus (1.5.50) | https://maxquant.net/perseus/ | N/A |
| Jupyter Notebook | https://jupyter.org/ | N/A |
| **Other** | | |
| Empore SPE SDB-RPS disk | Sigma Aldrich/Merck | Cat # 66886-U |
| 96-Well Plates | Thermo Fisher | Cat # AB-1300 |
| Silicone sealing mat, for 96 well PCR-plates | Nerbe Plus | Cat # 04-090-0000 |
| Reprosil-Pur Basic C18, 1.9 μm | Dr. Maisch Gmbh | Cat # r119.b9 |
| PicoFrit self-pack columns | Pico FRIT | Cat # PF360-75-15-N-5 |
| iST' sample preparation kit | PreOmics GmbH | Cat # P.O. 00001 |
| ThermoMixer® | Eppendorf | Cat # 460-0223 |
| Bravo Automated Liquid Handling Platform | Agilent | Cat # G5409A |
| NanoDrop™ One/OneC Microvolume UV-Vis Spectrophotometer | Thermo Fisher | Cat # ND-ONEC-W |
| Concentrator plus | Eppendorf | Cat # F-45-48-11 |
| EASY-nLC™ 1200 System | Thermo Fisher | Cat # LC140 |
| Q Exactive™ HF Hybrid Quadrupole-Orbitrap™ Mass Spectrometer | Thermo Fisher | Cat # IQLAAEGAAPFALGMBFZ |
| Q Exactive™ HF-X Hybrid Quadrupole-Orbitrap™ Mass Spectrometer | Thermo Fisher | Cat # 0726042 |

## Ethical approval

The study protocol was approved by the scientific-ethical committee of the Capital region of Denmark (H-1-2011-082) and registered with the Danish Data Protection Agency (2011-41-6410) and ClinicalTrials.gov (reg. no. NCT01492283). The study was conducted according to the principles of the Declaration of Helsinki, and oral and written informed consent was obtained from all participants.

## Study design

Plasma samples, all taken in the fasting state, were derived from two previously published studies. The details including corresponding

clinical data and laboratory data are in Junker *et al* (2016, 2015), and the groups are briefly described below. Patients had I) normal glucose tolerance (NGT) and no liver disease, II) NGT and NAFLD, III) T2D without liver disease, IV) T2D with NAFLD, and V) cirrhosis. NAFLD was diagnosed based on histology and graded according to hepatic fat infiltration: no NAFLD (< 5% fat infiltration), mild (5–33%), moderate (33–66%), and severe (> 66%). Non-alcoholic steatohepatitis (NASH) and fibrosis were graded according to the NAFLD activity score. Eight of the 20 patients with NAFLD also had NASH/fibrosis (all below fibrosis score 5). Cirrhosis was diagnosed histologically and clinically, based on signs of decompensation (e.g., ascites).

Individuals with T2D were diagnosed according to World Health Organization criteria. Exclusion criteria included weekly alcohol consumption of more than seven units for women and 14 units for men, treatment with steatogenic drugs within 3 months prior to inclusion, anemia, inflammatory bowel disease, gut resection, increased creatinine (> 150 μmol/l), albuminuria, or other chronic diseases. Controls were healthy (and matched with age, BMI, and gender) with no family history of diabetes, signs of liver disease (based on patient history, biochemical measurements, and ultrasound assessment), or other chronic diseases.

## Mouse experiments

To induce substantial obesity and NAFLD, we fed 8-week-old male C57Bl6/J mice (Charles River, River Laboratories, Wilmington, MA, USA) with a high-fat, high-sugar diet (HFD) comprising 58% kcal from fat (D12331; Research Diets, New Brunswick, NJ, USA) for 32 weeks. Another 6 male C57Bl6/J mice were maintained on regular chow diet and switched to HFD for 5 weeks at an age of 34 weeks to induce mild obesity and NAFLD. Mice were single- or double-housed on a 12:12-h light–dark cycle at 22°C with free access to water and food. Sample size estimation was done the same as in the human study (see Study design).

## Mouse pharmacology

The synthesis, purification, and characterization of the GLP-1/GIP co-agonist and the single GLP-1 and GIP mono-agonist controls were described previously, and the peptides were used without any further chemical modification or change in formulation (Finan *et al*, 2013). Whole-body composition (fat and lean mass) was measured with nuclear magnetic resonance technology (EchoMRI, Houston, TX, USA). For the treatment study in mice with severe NAFLD, DIO mice maintained for 32 weeks on HFD and then randomized to either vehicle, GLP-1, GIP, or GLP-1/GIP treatment according to body weight and body composition. Mice with mild NAFLD were fed HFD for 5 weeks and then treated with vehicle for the same time period as the other mouse groups. Compounds were administered in a vehicle of PBS (Gibco) and were given by daily subcutaneous injections at a dose of 10 nmol/kg at a volume of 5 μl per g body weight for 15 days. The investigators were not blinded to group allocation during the *in vivo* experiment. All procedures were approved by the Animal Use and Care Committee of Bavaria, Germany, in accordance with the Guide for the Care and Use of Laboratory Animals.

## Methods and Protocols

### Plasma sample preparation

All plasma samples were prepared according to the previously published plasma proteome profiling pipeline on an automated liquid handling system (Agilent Bravo) in a 96-well plate format (Geyer *et al*, 2016a). In brief, proteins were denatured, reduced, alkylated, and digested and peptides purified on StageTips (Kulak *et al*, 2014) using reagents from the PreOmics "iST" Kit (P.O. 00001, PreOmics GmbH).

In detail:

1  Transfer of 5 μl of blood plasma sample into an Eppendorf 96-well plate at 4°C on an Agilent Bravo liquid handling system (Plasma plate).
2  Make a 1:10 dilution by adding 45 μl of SDC reduction and alkylation buffer (included in PreOmics "iST" kit) into each well of the Plasma plate, mix thoroughly by pipetting 50 times up and down for a volume of 40 μl, and centrifuge the plate up to $300 \times g$.
3  Pipet 20 μl of tenfold diluted plasma into a new 96-well plate (Digestion plate).
4  Heat the plate at 95°C for 10 min.
5  Move the Digestion plate to room temperature and cool it down for 5 min. Meanwhile, prepare fresh trypsin/LysC mix in 0.05 (μg/μl) (total volume calculated by 20 μl per sample, 1:100 micrograms of enzyme to micrograms of protein).
6  Add 20 μl of trypsin/LysC mix into each well to a final volume of 40 μl.
7  Heat the Digestion plate at 37°C for 4 h (enzymatic digestion).
8  Quench the reaction by adding 40 μl of the PreOmics washing buffer 1, and mix thoroughly by pipetting 20 times up and down.
9  Prepare 96 StageTips on a home-made 3D-printed centrifugation block (2-plug SDB-RPS material (thickness $0.5 \pm 0.05$ mm) in 14-gauge).
10  Transfer 24 μl of the mixture onto the StageTips. Centrifuge the StageTip block at $1,500 \times g$ for 15 min.
11  Add 150 μl of the PreOmics washing buffer 1 to the StageTips, and centrifuge at $1,500 \times g$ for 15 min.
12  Add 150 μl of the PreOmics washing buffer 2 to the StageTips, and centrifuge at $1,500 \times g$ for 15 min.
13  Elute the peptides by adding 60 μl of elution buffer (1% ammonia in 80% acetonitrile), centrifuge at $1,500 \times g$ for 15 min, and collect peptides in PCR tube-stripes (0.2 ml volume).
14  Concentrate the peptide mixture by Speed-Vac at 60°C under vacuum for 90 min.
15  Re-suspend the peptide mixture in 40 μl of buffer A* (2% acetonitrile, 0.1% TFA in $ddH_2O$).

### High-pressure liquid chromatography and mass spectrometry

Samples were measured using LC-MS instrumentation consisting of an EASY-nLC 1200 system coupled to a nano-electrospray ion source and a Q Exactive HF Orbitrap (all Thermo Fisher Scientific). Purified peptides were separated on 40-cm HPLC columns (ID: 75 μm; in-house packed into the tip with ReproSil-Pur C18-AQ 1.9 μm resin (Dr. Maisch GmbH)). For each LC-MS/MS analysis, around 0.5 μg peptides was injected for the 45-min gradients and 1 μg for the fractions of the deep plasma dataset.

Additionally, we established very deep plasma proteome libraries. We pooled samples from all healthy individuals, NAFLD

patients, and liver cirrhosis patients separately and depleted the 14 highest abundant plasma proteins by serial depletion with a top6 (Multiple Affinity Removal Column Human 6; Agilent) and top14 depletion kits (Seppro Protein Depletion; Sigma-Aldrich). After digestion, the peptides were separated by the Spider Fractionator (Kulak *et al*, 2017) into 24 fractions. This library was combined with an additional peptide library that had been established in the same way in a separate study (Wewer Albrechtsen *et al*, 2018).

Peptides were loaded in buffer A (0.1% formic acid, 5% DMSO (v/v)) and eluted with a linear 35-min gradient of 3–30% of buffer B (0.1% formic acid, 5% DMSO, 80% (v/v) acetonitrile), followed by a 7-min increase to 75% of buffer B and a 1-min increase to 98% of buffer B, and a 2-min wash of 98% buffer B at a flow rate of 450 nl/min. Column temperature was kept at 60°C by a Peltier element containing an in-house-developed oven. For human plasma, MS spectra were acquired with a Top15 data-dependent MS/MS scan method (topN method) for the library and with the BoxCar scan method for study samples (Meier *et al*, 2018). The target value for the full scan MS spectra was $3 \times 10^6$ charges in the 300–1,650 *m/z* range with a maximum injection time of 55 ms and a resolution of 120,000 at *m/z* 200. Fragmentation of precursor ions was performed by higher-energy collisional dissociation (HCD) with a normalized collision energy of 27 eV. MS/MS scans were performed at a resolution of 15,000 at *m/z* 200 with an automatic gain control (AGC) target value of $5 \times 10^4$ and a maximum injection time of 25 ms. For mouse plasma, MS spectra were acquired with a data-independent acquisition (DIA) method. The DIA-MS method consisted of an MS1 scan from 350 to 1,650 *m/z* range (AGC target of $3 \times 10^6$, maximum injection time of 50 ms) at a resolution of 120,000 and 22 DIA segments (AGC target of $3 \times 10^6$, maximum injection time of 54 ms) at a resolution of 30,000 (Dataset EV3). Normalized stepped collision energy was set to 25, 27.5, 30, with a default charge state of 2. The acquisition of samples was randomized to avoid bias.

### Data analysis

For human plasma, mass spectrometric raw files were analyzed in the MaxQuant environment v.1.5.8.6 (Cox & Mann, 2008) employing the Andromeda search engine (Cox *et al*, 2011). The MS/MS spectra were searched against the human UniProt FASTA database (version 201704, 157,510 entries). Enzyme specificity was set to trypsin with a maximum of 2 missed cleavages, and the search included cysteine carbamidomethylation as fixed modification and oxidation on methionine and N-terminal acetylation as variable modifications with a minimum required peptide length of 7 amino acids. A false discovery rate (FDR) of 0.01 was set at the peptide and protein levels. Study samples were analyzed together with our in-house generated matching library and the "match between runs" algorithm (Nagaraj *et al*, 2012). Label-free quantitation (LFQ) was performed with a minimum ratio count of 2 and normalized in a separate group from the library raw files (Cox *et al*, 2014).

After filtering for "reverse", "only identified by site", "contaminants", and at least two valid values in any of the three technical replicates, we came to a dataset containing 684 protein groups (Dataset EV1, tab 2). We took the median value from the technical triplicates and further filtered the dataset for 70% data completeness in at least one experimental group. This resulted in 520 protein groups with 11% missing values (Dataset EV1, tab 3). We then replaced the missing values by drawing random samples from a

normal distribution (downshifted mean by 1.8 standard deviation (SD) and scaled SD (0.3) relative to that of proteome abundance distribution). This further resulted in a final dataset (Dataset EV1, tab 4) with which we performed the statistical analysis (except for global correlation analysis, which we filtered for 70% data completeness across all samples instead of one sub-group, resulting in 431 protein groups, without imputation (Dataset EV1, tab 5)).

For mouse plasma, DIA raw data were analyzed with Spectronaut Pulsar X™ with an in-house generated spectra library of mouse plasma that contains 8,899 peptides and 1,458 protein groups (mouse FASTA UniProt FASTA database version 201806 containing 92,096 entries). Spectronaut Pulsar X™ was used with default settings for DIA data with the decoy generation set to "mutated".

### Bioinformatic analysis

Bioinformatic analysis was performed in the Perseus platform (Tyanova *et al*, 2016) and Python scripts. Two-sample Student's *t*-test was used to determine the significantly changed proteins between disease and control groups with a permutation-based FDR of 0.05. For significant hits, minimal fold changes together with *P*-values (controlled by the s0 parameter in Perseus) were used with a permutation-based FDR of 0.05 resulting from an s0 of 0.01. For mouse plasma analysis, we used scipy.stats.ttest_ind to calculate *t*-test probabilities for the means of two independent samples. This was corrected for an FDR of 0.05 by the Benjamini–Hochberg method. Results were filtered to have both a significant FDR-corrected *P*-value and a minimum log2-fold change of ± 1.

Shapiro–Wilk test was applied to test normality of each individual protein across all five groups in the human study, and 75–80% of proteins are normally distributed. Levene's test was used to assess the equality of variances for each individual protein calculated for two groups in each of the NAFLD sub-cohorts. As a result, 95% of the proteins have equal variances. All six proteins reported in this study associated with NAFLD met the assumptions of the tests.

Liver-specific proteins were annotated according to the Human Protein Atlas, which defines "liver enriched", "group enriched", and "liver enhanced" proteins with at least 500% higher mRNA levels in liver compared to all other tissues, at least 500% higher mRNA levels in a group of 2–7 tissues compared to the rest, and at least 500% higher mRNA levels in the liver compared to average levels in all tissues, respectively.

Fold changes between conditions were calculated as Condition A/Condition B -1 for protein label-free (LFQ) intensities. The changes are indicated in percentage (e.g., increased by 62% and decreased by 27%).

## Data availability

The datasets in this study are available in the following databases:

- Proteomic dataset for the human cohorts: PRIDE archive PXD011839 (Vizcaino *et al*, 2016) (https://www.ebi.ac.uk/pride/archive/).
- Proteomic dataset for the mouse model: PRIDE archive PXD012056.

**Expanded View** for this article is available online.

## Acknowledgements

We thank all members of the Proteomics and Signal Transduction Group (Max Planck Institute) and the Clinical Proteomics group, in particular Atul Deshmukh for his help and discussions and Igor Paron, Christian Deiml, Korbinian Mayr, Gaby Sowa, Jeppe Madsen, Martin Rykær, and Simone Schopper for technical assistance. The work carried out in this project was partially supported by the Challenge Programme *MicrobLiver* funded by Novo Nordisk Foundation (Grant No. NNF15OC0016692), the Max Planck Society for the Advancement of Science, the European Union's Horizon 2020 research and innovation program (Grant Agreement No. 686547: MSmed project), the Novo Nordisk Foundation for the Clinical Proteomics group (Grant NNF15CC0001), the Novo Nordisk Foundation for the Copenhagen Bioscience PhD Program (NNF16CC0020906), and the Deutsche Forschungsgemeinschaft (DFG: SFB1123-A4).

## Author contributions

LN and PEG designed the study and performed and interpreted the MS-based proteomic analysis of plasma samples of the cohort, wrote the paper, and generated the figures. NJWA designed the study and contributed to the analysis and writing of the paper. PVT performed MS-based proteomic analysis of plasma samples and wrote the paper. SS administered co-agonist treatment and performed blood sampling and body weight measurements in mouse cohorts. JJH, FKK, TV, LLG, and AJ provided patient material and clinical data and revised the manuscript. AS and SD revised the manuscript. MHT, KS, SMH, and TDM designed and oversaw mouse experiment, provided mouse material, and revised the manuscript. MM designed and interpreted the MS-based proteomic analysis of patient plasma, supervised and guided the project, and wrote the paper.

## Conflict of interest

The authors declare that they have no conflict of interest.

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
