## [Review Process File · Molecular Systems Biology]

Plasma proteome profiling discovers novel proteins associated with non-alcoholic fatty liver disease

Lili Niu, Philipp E. Geyer, Nicolai J. Wewer Albrechtsen, Lise L. Gluud, Alberto Santos, Sophia Doll, Peter V. Treit, Jens J. Holst, Filip K. Knop, Tina Vilsbøll, Anders Junker, Stephan Sachs, Kerstin Stemmer, Timo D. Müller, Matthias H. Tschöp, Susanna M. Hofmann and Matthias Mann.

Review timeline:	Submission date:	12 th December 2018
	Editorial Decision:	10 th January 2019
	Revision received:	23 rd January 2019
	Editorial Decision:	25 th January 2019
	Revision received:	27 th January 2019
	Accepted:	28 th January 2019

Editor: Maria Polychronidou

Transaction Report:

1st Editorial Decision

10th January 2019

Thank you again for submitting your work to Molecular Systems Biology. We have now heard back from the three referees who agreed to evaluate your study. As you will see below, the reviewers acknowledge that the presented findings seem interesting. They raise however a series of concerns, which we would ask you to address in a major revision.

I think that the recommendations of the reviewers are clear and there is therefore no need to repeat the points listed below. As you may already know, our editorial policy allows in principle a single round of major revision so it is essential to provide responses to the reviewers' comments that are as complete as possible. Please feel free to contact me in case you would like to discuss in further detail any of the issues raised by the reviewers.

REFeree REPORTS

Reviewer #1:

The study by Niu et al aims to identify novel biomarkers of non-alcoholic fatty liver disease (NAFLD), a widespread condition that can progress to liver cirrhosis, necessitates invasive diagnostic procedure via biopsy, and currently lacks means of early detection. The authors examined levels of plasma proteins in 48 individuals, grouped into five cohorts according to their liver disease and type 2 diabetes statuses. They reported a number of proteins, abundance of which was significantly altered in individuals with NAFLD and cirrhosis, and speculated about possible biological significance of these changes based on what is known about these proteins in the literature. These findings were additionally tested using the mouse model of NAFLD/ cirrhosis and its current clinical treatment.

The study builds upon the previous methods developed in the PI's group - Plasma Proteome

Profiling and Boxcar - and applies them to the specific question of NAFLD diagnostics. As such, the work is somewhat weak in technical and conceptual innovation and is more of a proof of concept. The number of individuals in each cohort (N=10) is too small, making the study statistics underpowered for confident biomarker detection. That said, proteomic studies involving human subjects and plasma analysis are very challenging and an accomplishment on their own. The list of identified proteins also could be a good departure point for future studies by biochemists or clinicians. Overall, this is a scientifically sound piece of research, and if the issues with statistics and method/ results reporting outlined below are successfully resolved upon major revisions, I'd recommend the manuscript for publication.

Major points:

- 1) Statistical significance between human cohorts reported in the first part of the study is calculated using Student t-test; however, the described experimental design, where multiple cohorts are compared to each other and the groups within and between the cohorts serve as controls, requires different and more advanced statistical analysis. This is particularly important because the individuals' diabetes status is included as a potential source of variance that needs to be accounted for. Student t test could be applied to simple comparisons, like the ones in Fig 2A or the mouse experiment, but is not sufficient to determine significance in the experiments similar to the ones in Fig. 3B. The experimental design is set up to compare three or more groups at a time that have co-variables, and using multiple Student t tests in such situation can result in a lot of false positives. To address this issue, the authors should redo the analyses using for example ANOVA (or another suitable method) to establish if levels of proteins are at all different between the cohorts, then follow with Tukey HSD test to pinpoint which specific groups are different.
- 2) The supplied supplemental data in Table 1 contains protein measurements performed on human plasma. There are >600 proteins in the table (the number that is not mentioned in any context in the manuscript), and >26% of the reported measurements are missing (designated with NaN or zero). Were these measurements imputed for the final analysis and how was the imputation done? The procedure is not currently described in the Methods; a detailed description should be added, and the dataset that was used to generate figures and perform final analyses in the publication, including the imputed values, should be made available. Additionally, using a dataset with so many missing values (more than 1/4!) is excessively liberal. The authors mention that the final protein list was filtered to include only those that were measured in 70% of the plasma samples. What percent of the measurements were imputed in this final dataset? This specific list should be also provided in the supplement.
- 3) The authors emphasize the potential utility of DDP4 and PIGR as novel biomarkers. However, the follow-up experiments in mice undermine these findings, as the levels of these two proteins exhibit no response to the applied NAFLD treatment with GIP and GLP-1. It'd be interesting to see the authors comment on this.
- 4) The study uses methods previously published on by the group and in their description simply cites the respective publications (Geyer et al 2016; Meier et al 2018, etc.). The cited manuscripts contain many figures and numerous descriptions of method development and optimization. Thus, more specifics of what has been done and how the methods were applied to this study need to be included, particularly regarding the Boxcar method, which is quite new and may still require evaluation.

Minor points:

- 1) Fig. 2B shows a heat map in which columns (human subjects) were grouped using hierarchical clustering. Since the data shown in the heat map were already filtered to only contain proteins that were significantly different between "cirrhosis" and "NAFLD" groups, such clustering is meaningless/ artificial and should be omitted.
- 2) The authors constantly switch between expressing quantitative comparisons in terms of fold change (e.g., 1.7- fold) and percent (e.g., down by 27%). This is somewhat confusing, particularly when appears in the same sentence ("This was the case for four of the eight proteins, which changed also in NAFLD: PIGR (2.97-fold), LGALS3BP (1.70-fold), FCGBP (62%), and APOM (21%, Fig S1B)."). For the ease of interpretation, the authors should standardized how they report the changes.
- 3) "Plasma Correlation Profiling does not rely on specific protein epitopes and can therefore be generically applied across species to determine to what degree the biological responses are similar between them." This is true of any mass spec/ proteomics method and overstates uniqueness/ power of Plasma Proteome Profiling.

Reviewer #2:

In this study, Niu et al have tried to (i) identify novel proteins associated with NAFLD, and (ii) assess the effect of liver cirrhosis on the plasma proteome, by analyzing blood samples of 48 human patients with their well-established streamlined proteomics workflow together with their novel LC-MS methodology in an unbiased and non-hypothesis driven manner. The results indicate changes in plasma proteome in NAFLD and cirrhosis in which most of the liver-associated proteins were downregulated, suggesting a reflection of impaired hepatic protein synthesis in these patients. By using a mouse model for NAFLD, the authors nicely confirmed their human data, and further evaluated the effect of incretin agonists (alone and in combination) on improving metabolic phenotype and changing the plasma proteome associated with NAFLD. Moreover, using an unbiased correlation analysis, the authors further revealed that four of the clinical relevant markers currently used to evaluate liver disease (ALT, AST, ALP, and GGT) in humans are clustered into a single group together with a panel of five proteins, which were revealed in their proteome assessment. Overall, these findings are novel and interesting, and support the use of their previously published 'Plasma Proteome Profiling' technology to assess a disease state as well as the efficacy of a treatment. These findings also re-confirmed other studies that previously revealed the relevance of various proteins to the development of NAFLD. The paper is well written, and the current data support the conclusions offered in the paper.

Having said that, there is only one major issue that slightly dampen my enthusiasm about this study at this point, and addressing it would greatly improve the manuscript and its conclusions. The authors clearly show that two proteins (PIGR and ALDOB) are significantly associated with NAFLD both in humans and mice. Both proteins are upregulated in NAFLD and tend to decrease following the administration of incretin agonists. Evaluating their specific role in hepatic steatosis and/or cirrhosis in mice (by knocking them out/down) would further highlight their direct role in the development these disease states along with their potential to serve as biomarkers.

Reviewer #3:

The manuscript by Niu et al describes plasma proteome changes occurring during different stages of liver disease such as non-alcoholic fatty liver disease (NAFLD) and cirrhosis. The study included patient material from two previously collected cohorts published by Junker et al 2015 and 2016. Plasma proteome profiling of the patient cohorts revealed 8 proteins that show statistically significant differences in the two NAFLD groups compared to the two control groups. The findings were further investigated in an animal model using mice with mild to severe NAFLD to confirm increased levels for two of the eight proteins that were statistically elevated in the human cohorts. Although the human cohort used in this study was relatively small and would benefit from an independent validation cohort, the use of an animal model adds confidence to the observed results. Overall the manuscript demonstrates plasma proteome changes in liver disease in both a human cohort and in an animal model and represents an important contribution to the understanding of plasma proteome dynamics as the liver is a major producer of plasma proteins. Still, there are comments that needs to be addressed to further improve the manuscript.

Comment 1:

The authors state that the first cohort consists of ten obese patients with NAFLD and normal glucose tolerance (NGT) (NAFLD-cohort 1) and a second cohort of ten participants with both NAFLD and T2D (NAFLD-cohort 2). However, the clinical parameters indicate that three of the four groups have a BMI >30. As the definition of obesity is a BMI >30, the three groups (T2D_NAFLD, NAFLD_ngt and ngt) are obese, the other group (T2D) is overweight according to current BMI definition. The authors should clarify in the text outlining of the study design that three of the four groups are obese, and that one group is overweight.

Comment 2:

The plasma proteome profiling resulted in the identification of 1700 plasma proteins. But the authors do not report how many of these proteins are identified with a single peptide. The authors should state how many of the proteins are single peptide hits and mark or highlight in the Figures which proteins (if any) that are significantly regulated and only identified with a single peptide.

Comment 3:

In the proteome comparison between cirrhosis patients and healthy controls, the authors find reduced levels of coagulation proteins and carrier proteins and state that these changes can be related to prolonged bleeding, increased thrombosis risk and dyslipidemia. However, the measurements of prolonged bleeding and dyslipidemia in these particular samples is not included in the manuscript, which raises the question if the measurements were in fact performed. The authors should clarify these statements in the manuscript.

Comment 4:

Some of the regulated proteins found in this study are already incorporated into existing biomarker blood tests such as the SteatoTest and FibroTest. In addition, there are several surrogate non-invasive techniques for the diagnosis of NASH, staging of fibrosis and for assessing NAFLD. Some of these non-invasive techniques such as clinical assessments, ultrasound and blood tests were made in the study by Juncker et al 2016. It is highly recommended that the authors compare the performance of the 8-protein panel with some of the established non-invasive techniques or the SteatoTest/FibroTest.

Comment 5:

One of the overall long-term goals in the manuscript, is to help to improve detection of NAFLD and to predict disease progression to cirrhosis. According to the materials and method section the authors state that the samples were derived from two previously published studies by Juncker et al 2015 and 2016. The paper by Juncker et al 2016 states that 4 patients out of the total 20 NAFLD patients actually had NASH and an additional four patients had fibrosis. If this is the case, this information should be added to the methods and material section. Furthermore, it is unclear why the authors don't use this information to investigate proteome differences between these two subgroups compared to the 12 remaining NAFLD patients. This is of particular relevance for PIGR that the authors claim to be a novel marker of disease progression. If PIGR is a valid disease progression marker, higher levels would be anticipated in the subgroup that have NASH. It would also be interesting to know if LGALS3BP, which the authors claim to be a marker for fibrosis, is increased in the four NAFLD patients with confirmed fibrosis.

Comment 6:

The fact that PIGR correlates to ALT, AST, ALP and GGT, which are known liver damage markers, raises the questions if PIGR is really a better predictor for NAFLD than the above-mentioned established liver enzyme tests. The authors should provide comparative boxplots for ALT, AST, ALP and GGT over the four patient groups to show the comparison between these liver enzymes and PIGR to discriminate the non-NAFLD and the NAFLD group.

Comment 7:

In Figure 4A the authors state that the heat map includes 521 proteome variables but the text states 431 proteome variables. These numbers needs to be revised.

Point-by-point answers

A Protein Marker Panel for NAFLD

Point-by-point answers of reviewer's comments

We thank the reviewers for the in-depth and insightful comments on our manuscript "**Plasma proteome profiling discovers novel proteins associated with non-alcoholic fatty liver disease**".

We appreciate the positive evaluation of our efforts to detect new biomarker for the detection of liver diseases, using our Plasma Proteome Profiling technology. As the reviewers point out, investigating the liver disease marker candidates from the human cohorts with a mouse model and evaluating them with the global correlation map over the 48 participants in an orthogonal analysis method, gives us confidence in the identified proteins.

We have now performed additional statistical tests (ANOVA and Tukey's HSD) to ensure the statistical significance of the marker panel across multiple comparisons. Moreover, we included new paragraphs describing the technological novelty. This consists of combining the BoxCar scans and large libraries, resulting in increased coverage of the plasma proteome. We added more details and a stepwise protocol in the material and methods section.

Furthermore, we have in the meantime published a different study in which we investigated plasma protein levels in morbid obese individuals undergoing bariatric surgery (Albrechtsen et al., Cell Systems 2018). Intriguingly, in this unrelated study of 175 plasma proteomes, many of the candidates identified here, correlated with the classical liver damage marker ALT, providing further evidence for their involvement in liver damage (as these are proteins originating from the liver). We believe that these data add to the manuscript and now we include an analysis in the supplementary material.

As the reviewers also point out, the novel candidates are well supported by multiple lines of evidence and are very likely to be liver damage markers. To our knowledge, this is one of the first times, if not the first time, that MS-based plasma proteomics has uncovered such biomarker candidates. With our clinical partners, we are now setting out to define in which specific liver disease these damage markers are involved in.

Finally, we have responded to all reviewers comments and changed the text accordingly.

Reviewer #1:

The study by Niu et al aims to identify novel biomarkers of non-alcoholic fatty liver disease (NAFLD), a widespread condition that can progress to liver cirrhosis, necessitates invasive diagnostic procedure via biopsy, and currently lacks means of early detection. The authors examined levels of plasma proteins in 48 individuals, grouped into five cohorts according to their liver disease and type 2 diabetes statuses. They reported a number of proteins, abundance of which was significantly altered in individuals with NAFLD and cirrhosis, and speculated about possible biological significance of these changes based on what is known about these proteins in the literature. These findings were additionally tested using the mouse model of NAFLD/ cirrhosis and its current clinical treatment.

The study builds upon the previous methods developed in the PI's group - Plasma Proteome Profiling and Boxcar - and applies them to the specific question of NAFLD diagnostics. As such, the work is somewhat weak in technical and conceptual innovation and is more of a proof of concept. The number of individuals in each cohort (N=10) is too small, making the study statistics underpowered for confident biomarker detection. That said, proteomic studies involving human subjects and plasma analysis are very challenging and an accomplishment on their own. The list of identified proteins also could be a good departure point for future studies by biochemists or clinicians. Overall, this is a scientifically sound piece of research, and if the issues with statistics and method/ results reporting outlined below are successfully resolved upon major revisions, I'd recommend the manuscript for publication.

Authors' response

We thank the reviewer for the constructive and in-depth analysis of our work. We agree that the paper is a step in our efforts to develop technologies, conceptual and clinical advances and apply them to clinically important problems. As such, not all of the technologies are novel, but we would like to point out that this is the first study defining biomarkers for liver damage and that this would not have been possible without the 'BoxCar' scan in plasma proteomics, which has just been published by us half a year ago. We also agree that the cohorts are relatively small. However, this is par for the course in the field, in which studies with about 10 cases and controls routinely are published. Here we aggregate three such studies, which allows us to validate across them, and additionally investigate a mouse model, all with novel plasma proteome technology.

Major points:

1. Statistical significance between human cohorts reported in the first part of the study is calculated using Student t-test; however, the described experimental design, where multiple cohorts are compared to each other and the groups within and between the cohorts serve as controls, requires different and more advanced statistical analysis. This is particularly important because the individuals' diabetes status is

included as a potential source of variance that needs to be accounted for. Student t test could be applied to simple comparisons, like the ones in Fig 2A or the mouse experiment, but is not sufficient to determine significance in the experiments similar to the ones in Fig. 3B. The experimental design is set up to compare three or more groups at a time that have co-variables, and using multiple Student t tests in such situation can result in a lot of false positives. To address this issue, the authors should redo the analyses using for example ANOVA (or another suitable method) to establish if levels of proteins are at all different between the cohorts, then follow with Tukey HSD test to pinpoint which specific groups are different.

The objective of this analysis was to identify differentially regulated proteins in one group vs its baseline and not a cross-comparisons between all disease states. However, we have now followed the reviewer's suggestion and performed the statistical analysis on all five experimental groups in the human cohort. A One-way ANOVA controlled for FDR below 0.05 by Benjamini/Hochberg, confirmed the six of the eight reported proteins as significantly different across the experimental groups, including ALDOB, PIGR, LGALS3BP, AFM, VTN and APOM (See tables below). Tukey's HSD test at $\alpha=0.05$ resulted in the specific groups that are different.

One-way ANOVA with corrected FDR (<0.05) by Benjamini/Hochberg

	F-value	pvalue	-Log pvalue	qvalue	rejected
ALDOB	9.88	0.0	5.03	0.0	True
PIGR	8.32	0.0	4.34	0.0	True
LGALS3BP	4.96	0.0	2.65	0.04	True
AFM	5.56	0.0	2.97	0.03	True
VTN	7.12	0.0	3.76	0.01	True
APOM	7.11	0.0	3.76	0.01	True
FCGBP	4.06	0.01	2.15	0.07	False
IGHD	2.5	0.06	1.25	0.22	False

Tukey's HSD test at $\alpha=0.05$

	group1	group2	log2FC	lower	upper	rejected
PIGR	Cirrhosis	T2D	-1.7	-3.06	-0.35	True
PIGR	Cirrhosis	ktrl	-2.07	-3.35	-0.79	True
PIGR	NAFLD_T2D	ktrl	-1.63	-2.91	-0.34	True
PIGR	NAFLD_ngt	T2D	-1.5	-2.86	-0.14	True
PIGR	NAFLD_ngt	ktrl	-1.86	-3.14	-0.58	True
ALDOB	Cirrhosis	NAFLD_T2D	3.0	1.33	4.67	True
ALDOB	Cirrhosis	NAFLD_ngt	2.07	0.4	3.73	True
ALDOB	NAFLD_T2D	ktrl	-2.94	-4.61	-1.27	True
ALDOB	NAFLD_ngt	ktrl	-2.0	-3.67	-0.33	True
LGALS3BP	Cirrhosis	T2D	-1.19	-2.08	-0.29	True
LGALS3BP	NAFLD_T2D	T2D	-1.04	-1.94	-0.15	True
VTN	Cirrhosis	NAFLD_T2D	0.22	0.02	0.42	True
VTN	NAFLD_T2D	ktrl	-0.31	-0.51	-0.12	True
VTN	NAFLD_ngt	ktrl	-0.28	-0.48	-0.08	True
VTN	T2D	ktrl	-0.24	-0.45	-0.03	True
AFM	Cirrhosis	NAFLD_T2D	0.39	0.01	0.78	True
AFM	NAFLD_T2D	T2D	-0.53	-0.94	-0.13	True
AFM	NAFLD_T2D	ktrl	-0.56	-0.94	-0.18	True
APOM	Cirrhosis	ktrl	0.29	0.0	0.57	True
APOM	NAFLD_T2D	NAFLD_ngt	0.37	0.09	0.66	True
APOM	NAFLD_T2D	T2D	0.42	0.12	0.73	True
APOM	NAFLD_T2D	ktrl	0.42	0.13	0.7	True

Figure 3. A panel of proteins strongly associated with NAFLD in the human cohorts.

2. The supplied supplemental data in Table 1 contains protein measurements performed on human plasma. There are >600 proteins in the table (the number that is not mentioned in any context in the manuscript), and >26% of the reported measurements are missing (designated with NaN or zero). Were these measurements imputed for the final analysis and how was the imputation done? The procedure is not currently described in the Methods; a detailed description should be added, and the dataset that was used to generate figures and perform final analyses in the publication, including the imputed values, should be

made available. Additionally, using a dataset with so many missing values (more than 1/4!) is excessively liberal. The authors mention that the final protein list was filtered to include only those that were measured in 70% of the plasma samples. What percent of the measurements were imputed in this final dataset? This specific list should be also provided in the supplement.

It was indeed not clearly explained, how we arrived at the final dataset. In the revised manuscript, we added details for the data processing in the "Materials and Method" session under "Data analysis" (paragraph 4, page 14). We also added the datasets at each step of the analysis pipeline in "Dataset EV1". In the Materials and Methods of the revised manuscript, we have added the following text:

"After filtering for "reverse", "only identified by site", "contaminants" and at least two valid values in any of the three technical replicates, the dataset contained in total 684 protein groups (Dataset EV1, tab2). We took the median value from the technical triplicates and further filtered the dataset for 70% data completeness in at least one experimental group. This resulted in 520 protein groups with 11% missing values (Dataset EV1, tab3). We then replaced the missing values by drawing random samples from a normal distribution (down-shifted mean by 1.8 standard deviation (SD) and scaled SD (0.3) relative to that of proteome abundance distribution. This further resulted in a final dataset (Dataset EV1, tab4) with which we performed the statistical analysis (except for global correlation analysis, which we filtered for 70% data completeness across all samples instead of one sub-group, resulting in 431 protein groups (Dataset EV1, tab5))."

In the global correlation analysis (Fig 4), there is no imputation because we only used proteins quantified in at least 70% of samples (431 proteins). This is sufficient for correlation analysis without imputation (supplemental table Dataset EV1, tab5). This is pointed out in the revised manuscript.

3. The authors emphasize the potential utility of DDP4 and PIGR as novel biomarkers. However, the follow-up experiments in mice undermine these findings, as the levels of these two proteins exhibit no response to the applied NAFLD treatment with GIP and GLP-1. It'd be interesting to see the authors comment on this.

In the mouse experiment, both Pigr and Dpp4 showed a decreasing but not significant trend when comparing the severe NAFLD group (HFD>6m) with the GLP-1/GIP co-agonist treatment group – the group with largest weight loss. Behavior of Pigr across the groups is very similar to three established inflammation marker – Saa1, Apcs and Orm2 (Figure 5G-I). Their levels also showed a decreasing trend. These three inflammation markers indicate that even though there is weight loss, there is still remaining inflammation. This is exactly the behavior that we have seen in human weight loss studies as well (Geyer et al., Mol Syst Biol, 2016; Albrechtsen et al., Cell Systems 2018). Moreover, the fact that plasma levels of PIGR are statistically significantly higher in both human NAFLD cohorts, and that its levels correlate with four clinically

assessed liver enzymes, independently validated in a mouse cohort. The fact that fold changes are even higher contribute to making PIGR a very interesting biomarker candidate.

4. The study uses methods previously published on by the group and in their description simply cites the respective publications (Geyer et al 2016; Meier et al 2018, etc.). The cited manuscripts contain many figures and numerous descriptions of method development and optimization. Thus, more specifics of what has been done and how the methods were applied to this study need to be included, particularly regarding the Boxcar method, which is quite new and may still require evaluation.

We use the latest technology that has just been published in Nature Methods half a year ago and this is the first application to a biomarker study. However, we agree with the reviewer that the reader would profit from more detailed description of the technological improvements. We have now added additional paragraphs to the results and the methods sections:

Result section (paragraph 3, page 4):

“We improved our previously described Plasma Proteome Profiling pipeline (Geyer et al., 2016a) by deep libraries in combination with a new acquisition method termed BoxCar (Meier et al., 2018). This method results in a ten-fold increased dynamic range of peptide signals due to equalized filling pattern of the orbitrap mass analyzer. The library consists of pooled and depleted plasma samples – including from patients with liver diseases (see Materials and methods) – that were separated at the peptide level into 24 fractions. This deep plasma library included in total 2081 proteins. The study samples were then measured without depletion, but with BoxCar scans and in technical triplicates, resulting in about 150 LC-MS/MS data sets (Fig 1B)...”

Method section (paragraph 1, page 13):

“Additionally, we established very deep plasma proteome libraries. We pooled samples from all healthy individuals, NAFLD patients and liver cirrhosis patients separately and depleted the 14 highest abundant plasma proteins by serial depletion with a top6 (Multiple Affinity Removal Column Human 6, Agilent) and top14 depletion kits (Seppro Protein Depletion, Sigma-Aldrich). After digestion, the peptides were separated by the Spider Fractionator (Kulak et al., 2017) into 24 fractions. This library was combined with an additional peptide library that had been established in the same way in a separate study (Albrechtsen et al., 2018).”

With respect to validation of the BoxCar method, it was extensively tested in the Nature Methods publication. Additionally, we have very recently published a plasma proteomics study on bariatric surgery,

in which the patients undergo extensive weight loss (Albrechtsen et al., Cell Systems 2018). In that study, we were able to confirm previous findings like inflammation proteins are decreasing due to the weight loss (Geyer et al., 2016b). As there are clear confirmatory results, from a biological point of view, we are confident with the applied method.

Minor points:

1. Fig. 2B shows a heat map in which columns (human subjects) were grouped using hierarchical clustering. Since the data shown in the heat map were already filtered to only contain proteins that were significantly different between "cirrhosis" and "NAFLD" groups, such clustering is meaningless/ artificial and should be omitted.

Indeed the proteins shown in Fig 2B were only the ones significantly different between "cirrhosis" and "non-NAFLD" group. We politely disagree that the clustering is meaningless. We used the heatmap as a visualization tool to show the levels of the individual proteins across the study participants. Moreover, the hierarchical clustering in combination with the color-coding of the proteins for different Gene Ontology Biological Process annotations visualizes that proteins of the inflammation system tend to be increased in cirrhosis patients whereas carrier proteins and proteins of the coagulation system are elevated in the non-NAFLD group.

2. The authors constantly switch between expressing quantitative comparisons in terms of fold change (e.g., 1.7- fold) and percent (e.g., down by 27%). This is somewhat confusing, particularly when appears in the same sentence ("This was the case for four of the eight proteins, which changed also in NAFLD: PIGR (2.97-fold), LGALS3BP (1.70-fold), FCGBP (62%), and APOM (21%, Fig S1B)."). For the ease of interpretation, the authors should standardized how they report the changes.

We did this because some of the fold changes are quite large and because it is awkward to express decreases as fold-changes. However, in line with the reviewer's suggestion and at the discretion of the editor, we have now standardized all fold changes in the form of percentage. Additionally, we added the following explanation to the manuscript in the "Materials and Method" session under "Bioinformatic analysis" (paragraph 5, page 14):

"Fold changes between conditions were calculated as Condition A / Condition B -1 for protein label-free (LFQ) intensities. The changes are indicated in percentage (e.g. increased by 62% and decreased by 27%). "

3. "Plasma Correlation Profiling does not rely on specific protein epitopes and can therefore be generically applied across species to determine to what degree the biological responses are similar between them."

This is true of any mass spec/ proteomics method and overstates uniqueness/ power of Plasma Proteome Profiling.

We completely agree with the reviewer that the sentence is true for all mass spectrometry-based proteomics methods and not unique for Plasma Proteome Profiling. Our intention was to distinguish between mass spectrometry-based proteomics and immune-assay-based platforms, as we have frequently seen that clinical collaboration partners do not appreciate this point. We now changed the sentence as follows (paragraph 5, page 7):

“As a MS-based method, Plasma Proteome Profiling does not rely on specific protein epitopes (unlike most antibodies) and can generically be applied across species to determine to which degree the biological responses are similar between them.”

Reviewer #2:

In this study, Niu et al have tried to (i) identify novel proteins associated with NAFLD, and (ii) assess the effect of liver cirrhosis on the plasma proteome, by analyzing blood samples of 48 human patients with their well-established streamlined proteomics workflow together with their novel LC-MS methodology in an unbiased and non-hypothesis driven manner. The results indicate changes in plasma proteome in NAFLD and cirrhosis in which most of the liver-associated proteins were downregulated, suggesting a reflection of impaired hepatic protein synthesis in these patients. By using a mouse model for NAFLD, the authors nicely confirmed their human data, and further evaluated the effect of incretin agonists (alone and in combination) on improving metabolic phenotype and changing the plasma proteome associated with NAFLD. Moreover, using an unbiased correlation analysis, the authors further revealed that four of the clinical relevant markers currently used to evaluate liver disease (ALT, AST, ALP, and GGT) in humans are clustered into a single group together with a panel of five proteins, which were revealed in their proteome assessment. Overall, these findings are novel and interesting, and support the use of their previously published 'Plasma Proteome Profiling' technology to assess a disease state as well as the efficacy of a treatment. There findings also re-confirmed other studies that previously revealed the relevance of various proteins to the development of NAFLD. The paper is well written, and the current data support the conclusions offered in the paper.

Having said that, there is only one major issue that slightly dampen my enthusiasm about this study at this point, and addressing it would greatly improve the manuscript and its conclusions. The authors clearly show that two proteins (PIGR and ALDOB) are significantly associated with NAFLD both in humans and mice. Both proteins are upregulated in NAFLD and tend to decrease following the administration of incretin agonists. Evaluating their specific role in hepatic steatosis and/or cirrhosis in mice (by knocking them

out/down) would further highlight their direct role in the development these disease states along with their potential to serve as biomarkers.

Authors' response:

We thank the reviewer for the positive and encouraging evaluation of our manuscript and technology, including the key novelties and main messages. Indeed, we are convinced that the combination of the global correlation analysis as an orthogonal method to the case vs control comparisons in combination with the mouse study provides much confidence in the identified protein panel.

We agree that identifying the biological role of PIGR and ALDOB as well as their involvement in the development of the different liver disease states would be highly valuable. However, we hope the reviewer agrees that adding an additional mouse study is out of the scope of this manuscript. Nevertheless, to further increase evidence for our protein candidates we now re-analyzed another Plasma Proteome Profiling manuscript, which has appeared in the meantime (Albrechtsen et al., Cell Systems 2018). In that study, we investigated the changes of the plasma proteome upon bariatric surgery in morbidly obese individuals. ALT levels were determined, providing a proxy for liver damage and gratifyingly PIGR, ALDOB and ANPEP correlated statistically significant with ALT. Moreover, four other proteins of the identified panel also correlated positively. We will supply these data as a supplemental figure (Fig S4) together with this additional text (paragraph 2, page 7):

“After identifying proteins correlating with the established liver damage marker ALT by the global correlation map, we asked, if we can reproduce these correlations in our very recently published study on the effect of bariatric surgery in morbidly obese individuals on the plasma proteome (Albrechtsen et al., 2018). Correlating protein levels to clinically determined ALT levels across 175 plasma samples confirmed that ALDOB, PIGR and ANPEP are statistically significantly correlated, whereas TGFBI is very close to the threshold (Fig S4).”

**Reviewer #3:**

The manuscript by Niu et al describes plasma proteome changes occurring during different stages of liver disease such as non-alcoholic fatty liver disease (NAFLD) and cirrhosis. The study included patient material from two previously collected cohorts published by Junker et al 2015 and 2016. Plasma proteome profiling of the patient cohorts revealed 8 proteins that show statistically significant differences in the two NAFLD groups compared to the two control groups. The findings were further investigated in an animal model using mice with mild to severe NAFLD to confirm increased levels for two of the eight proteins that were statistically elevated in the human cohorts. Although the human cohort used in this study was relatively small and would benefit from an independent validation cohort, the use of an animal model adds confidence to the observed results. Overall the manuscript demonstrates plasma proteome changes in liver disease in both a human cohort and in an animal model and represents an important contribution to the understanding of plasma proteome dynamics as the liver is a major producer of plasma proteins. Still, there are comments that needs to be addressed to further improve the manuscript.

Authors' response:

We thank the reviewer for the positive evaluation of our work, including the appreciation of the mouse model as a confirmatory aspect.

1. The authors state that the first cohort consists of ten obese patients with NAFLD and normal glucose tolerance (NGT) (NAFLD-cohort 1) and a second cohort of ten participants with both NAFLD and T2D (NAFLD-cohort 2). However, the clinical parameters indicate that three of the four groups have a BMI >30. As the definition of obesity is a BMI >30, the three groups (T2D_NAFLD, NAFLD_ngt and ngt) are obese, the other group (T2D) is overweight according to current BMI definition. The authors should clarify in the text outlining of the study design that three of the four groups are obese, and that one group is overweight.

Matching patients in all aspects across different diseases (here liver and diabetes) is challenging, especially across different studies. However there is no significant difference between the groups within either of the two studies the median was 27kg/m² and the interquartile range was 24-36, as previously pointed out for the underlying study (Junker et al., J Intern Med, 2016; Junker et al., J Gastroenterol Hepatol, 2015).

2. The plasma proteome profiling resulted in the identification of 1700 plasma proteins. But the authors do not report how many of these proteins are identified with a single peptide. The authors should state how many of the proteins are single peptide hits and mark or highlight in the Figures which proteins (if any) that are significantly regulated and only identified with a single peptide.

We thank the reviewer for pointing this out. In the revised manuscript, we now analyze the results of a study published in the meantime (Albrechtsen et al., Cell Systems 2018) together with our data. In particular, this expands the matching library, which now encompasses 72 fractionated plasma runs (24 fractions x 3 pooled samples representing non-NAFLD, NAFLD and cirrhosis groups). With this enlarged library, we identified 2081 proteins in total (252 by single peptide identifications). We updated the numbers in the manuscript (paragraph 3, page 4).

3. In the proteome comparison between cirrhosis patients and healthy controls, the authors find reduced levels of coagulation proteins and carrier proteins and state that these changes can be related to prolonged bleeding, increased thrombosis risk and dyslipidemia. However, the measurements of prolonged bleeding and dyslipidemia in these particular samples is not included in the manuscript, which raises the question if the measurements were in fact performed. The authors should clarify these statements in the manuscript.

We clarified this in the revised manuscript. We followed the guidelines of the American Association for the Study of Liver Diseases and the National Guidelines on diagnosis and monitoring of cirrhosis, which includes regular assessments of the coagulation system (INR, factor II, VII, X, Thrombocytes etc.).

4. Some of the regulated proteins found in this study are already incorporated into existing biomarker blood tests such as the SteatoTest and FibroTest. In addition, there are several surrogate non-invasive techniques for the diagnosis of NASH, staging of fibrosis and for assessing NAFLD. Some of these non-invasive techniques such as clinical assessments, ultrasound and blood tests were made in the study by

Juncker et al 2016. It is highly recommended that the authors compare the performance of the 8-protein panel with some of the established non-invasive techniques or the SteatoTest/FibroTest.

A direct comparison of our protein panel to established non-invasive techniques and other tests would be highly valuable and indeed, this will be the next step in our endeavor to bring this panel to clinical application. For this purpose, we are in an international-consortium called Microb-Liver, running for another four years. At the moment, new clinical cohorts are being established with in-depth phenotyping of patients with different kinds of liver diseases. Our aim is to confirm the herein reported liver marker panel and compare their performance to already established non-invasive, but also invasive tests. This involves a dedicated Ph.D. student in the collaborating clinical group and should be finished in about three years. In the meantime, the community will be able to assess our biomarker candidates as well.

5. One of the overall long-term goals in the manuscript, is to help to improve detection of NAFLD and to predict disease progression to cirrhosis. According to the materials and method section the authors state that the samples were derived from two previously published studies by Junker et al 2015 and 2016. The paper by Junker et al 2016 states that 4 patients out of the total 20 NAFLD patients actually had NASH and an additional four patients had fibrosis. If this is the case, this information should be added to the methods and material section. Furthermore, it is unclear why the authors don't use this information to investigate proteome differences between these two subgroups compared to the 12 remaining NAFLD patients. This is of particular relevance for PIGR that the authors claim to be a novel marker of disease progression. If PIGR is a valid disease progression marker, higher levels would be anticipated in the subgroup that have NASH. It would also be interesting to know if LGALS3BP, which the authors claim to be a marker for fibrosis, is increased in the four NAFLD patients with confirmed fibrosis.

We have now added the information on the eight individuals that had NASH/fibrosis to the materials and methods in the revised manuscript (paragraph 2, page 11). Neither of them had a fibrosis score over 5 (on a scale up to 10). All four who had NASH were the same who had fibrosis. Below we show the plasma levels of PIGR and LGALS3BP across the range of fibrosis scores. Due to mild fibrosis and also limited sample size in each group, we cannot yet assign our candidates to specific liver disease subtypes, but we plan to address this in future studies (see above).

Fibrosis histology score	0	1	2	4
N	11	5	1	1

6. The fact that PIGR correlates to ALT, AST, ALP and GGT, which are known liver damage markers, raises the questions if PIGR is really a better predictor for NAFLD than the above-mentioned established liver enzyme tests. The authors should provide comparative boxplots for ALT, AST, ALP and GGT over the four patient groups to show the comparison between these liver enzymes and PIGR to discriminate the non-NAFLD and the NAFLD group.

We performed a One-way ANOVA test on four liver enzymes and PIGR across all five groups in the human cohort. The result showed that PIGR and three liver enzymes – ALT, AST, and GGT – are significantly different across groups. Followed by a Tukey’s HSD test (a more stringent test than students’ t-test which we used in Fig 3B, as here we are comparing across different groups, a purpose different from Fig 3B), PIGR was significantly different in 5 pair-wise comparisons and showed higher practical significance (effect size/fold change). However, ALT is not able to discriminate between the non-NAFLD groups and cirrhosis group while PIGR clearly can. The fold change is even higher in the cirrhosis group. This clearly indicates that PIGR has great potential to be included in a liver damage test panel. We plan to examine this in larger cohorts in the future (see above).

One-way ANOVA followed by Tukey's HSD test at $\alpha=0.05$

	group1	group2	log2FC	lower	upper	rejected
PIGR	Cirrhosis	T2D	-1.7	-3.06	-0.35	True
PIGR	Cirrhosis	ktrl	-2.07	-3.35	-0.79	True
PIGR	NAFLD_T2D	ktrl	-1.62	-2.91	-0.34	True
PIGR	NAFLD_ngt	T2D	-1.5	-2.86	-0.14	True
PIGR	NAFLD_ngt	ktrl	-1.86	-3.14	-0.58	True
ALT	Cirrhosis	NAFLD_T2D	1.39	0.5	2.27	True
ALT	Cirrhosis	NAFLD_ngt	1.2	0.32	2.09	True
ALT	NAFLD_T2D	T2D	-1.52	-2.5	-0.55	True
ALT	NAFLD_T2D	ktrl	-1.89	-2.77	-1.0	True
ALT	NAFLD_ngt	T2D	-1.34	-2.32	-0.36	True
ALT	NAFLD_ngt	ktrl	-1.7	-2.59	-0.82	True
AST	NAFLD_T2D	ktrl	-0.95	-1.74	-0.16	True
AST	NAFLD_ngt	ktrl	-1.04	-1.83	-0.25	True
GGT	Cirrhosis	ktrl	-1.72	-3.39	-0.05	True
GGT	NAFLD_ngt	ktrl	-2.22	-3.9	-0.55	True

7. In Figure 4A the authors state that the heat map includes 521 proteome variables but the text states 431 proteome variables. These numbers needs to be revised.

We highly appreciate that the reviewer spotted this mistake, which we have corrected now. For the imputation and correlation analysis, please also see the answer to reviewer #1, major point 2.

2nd Editorial Decision

25th January 2019

Thank you again for sending us your revised manuscript. We are satisfied with the performed revisions and we think that the study is now suitable for publication.

Corresponding Author Name: Matthias Mann

Manuscript Number: MSB-18-8793